# Intestinal infection regulates behavior and learning via neuroendocrine signaling

Jogender Singh, Alejandro Aballay*

Department of Molecular Microbiology & Immunology, Oregon Health & Science University, Portland, United States

**Abstract** The recognition of pathogens and subsequent activation of defense responses are critical for the survival of organisms. The nematode *Caenorhabditis elegans* recognizes pathogenic bacteria and elicits defense responses by activating immune pathways and pathogen avoidance. Here we show that chemosensation of phenazines produced by pathogenic *Pseudomonas aeruginosa*, which leads to rapid activation of DAF-7/TGF-β in ASJ neurons, is insufficient for the elicitation of pathogen avoidance behavior. Instead, intestinal infection and bloating of the lumen, which depend on the virulence of *P. aeruginosa*, regulates both pathogen avoidance and aversive learning by modulating not only the DAF-7/TGF-β pathway but also the G-protein coupled receptor NPR-1 pathway, which also controls aerotaxis behavior. Modulation of these neuroendocrine pathways by intestinal infection serves as a systemic feedback that enables animals to avoid virulent bacteria. These results reveal how feedback from the intestine during infection can modulate the behavior, learning, and microbial perception of the host.

*For correspondence:
aballay@ohsu.edu

Competing interests: The authors declare that no competing interests exist.

## Introduction

In nature, all organisms are continuously exposed to a complex environment and are under constant threat of attack by pathogenic microbes. To ensure their survival, organisms must recognize pathogens and mount a robust defense in response to their attack. Physical avoidance of pathogens is also a critical defense strategy to reduce pathogen infections (*Kavaliers et al., 2019*; *Medzhitov et al., 2012*). Chemosensation of bacterial metabolites and toxins appears to play an important role in eliciting pathogen avoidance behaviors. In mammals, olfactory chemosensory neurons and nociceptor sensory neurons detect bacterial toxins, quorum-sensing molecules, formyl peptides, and lipopolysaccharides through distinct molecular mechanisms that lead to rapid avoidance behaviors (*Boillat et al., 2015*; *Chiu et al., 2013*; *Meseguer et al., 2014*; *Rivière et al., 2009*; *Tizzano et al., 2010*; *Yang and Chiu, 2017*). Similarly, in the fruit fly *Drosophila melanogaster*, olfactory and gustatory neurons have been reported to detect geosmin (the smell of mold), phenol, and lipopolysaccharides via distinct molecular mechanisms, allowing the organism to avoid food contaminated with bacteria (*Mansourian et al., 2016*; *Soldano et al., 2016*; *Stensmyr et al., 2012*). A deeper understanding of the various mechanisms of pathogen avoidance has the potential to uncover conserved host defense responses that are important against pathogen infections.

The free-living nematode *Caenorhabditis elegans* feeds on bacteria. Pathogenic bacteria such as *Pseudomonas aeruginosa* infect and kill *C. elegans*, and upon exposure to *P. aeruginosa*, *C. elegans* elicits an innate immune response that results in the activation of microbial-killing pathways and a pathogen-avoidance behavior that improve its survival (*Martin et al., 2017*; *Meisel and Kim, 2014*; *Styer et al., 2008*; *Troemel et al., 2006*). Chemosensation of the *P. aeruginosa* metabolites phenazine-1-carboxamide and pyochelin has been proposed to be responsible for eliciting the avoidance behavior (*Meisel et al., 2014*). These metabolites activate a G-protein signaling pathway in the ASJ chemosensory neuron pair that induces expression of the neuromodulator DAF-7/TGF-β within minutes of exposure to the pathogen. However, the animals are initially attracted towards lawns of

**eLife digest** The bacteria that cause disease may be microscopic, but animals can use senses other than sight to protect themselves from infection. Some bacteria produce harmful toxins, which animals can instinctively recognize as being dangerous using their sense of smell or taste. This is called chemosensation, an innate ability that allows animals to react to chemical stimuli.

But animals can also 'learn' to avoid harmful bacteria, though it remains unclear how they do so. Now, Singh and Aballay have used roundworms as a model organism to study this phenomenon.

Roundworms feed on bacteria, so they need to be able to distinguish between disease-causing strains and harmless ones. However, they only start avoiding harmful bacteria after several hours of exposure, which would not necessarily be expected if they were using chemosensation. This prompted Singh and Aballay to investigate whether another mechanism could be teaching the roundworms to avoid disease-causing bacteria, by comparing roundworms that had been exposed to harmful or benign strains.

As observed previously, the roundworms learned to avoid the harmful bacterium *Pseudomonas aeruginosa*. However, exposing the worms to certain chemicals produced by *P. aeruginosa* was not enough to teach them to avoid the bacterium. Instead, the experiments showed that when roundworms ingested disease-causing bacteria, the infection caused intestinal bloating. The more toxic the bacteria, the more the intestine swelled, triggering a neural pathway associated with a preference for oxygen. In a few hours, the worms learned to avoid the low oxygen environment associated with *P. aeruginosa* and developed a preference for high oxygen conditions surrounding harmless bacteria such as *Escherichia coli*.

These results show how an intestinal infection can send signals to the nervous system to modulate animal behavior. Moreover, Singh and Aballay have identified a neural pathway that stimulates a behavioral host response to defend against infection.

*P. aeruginosa* and only begin to avoid the pathogen after exposure for hours (*Meisel et al., 2014*; *Singh and Aballay, 2019a*; *Sun et al., 2011*). The initial attraction of *C. elegans* towards *P. aeruginosa* is mediated by sensing of quorum-sensing molecules acylated homoserine lactones (*Beale et al., 2006*; *Ha et al., 2010*). In addition, in a two-choice preference assay between *Escherichia coli* and *P. aeruginosa*, the naïve animals show a higher preference for *P. aeruginosa* (*Ha et al., 2010*; *Zhang et al., 2005*). The animals only change their choice and move towards *E. coli* lawns after an exposure time to the two bacteria exceeding 4 hr (*Zhang et al., 2005*). Thus, the activation of DAF-7/TGF-β signaling in ASJ neurons, which takes place after minutes of exposure to *P. aeruginosa*, does not fully explain the later aversive learning towards *P. aeruginosa*.

Here we show that chemosensation of *P. aeruginosa* metabolites, which leads to the induction of DAF-7/TGF-β in ASJ neurons, does not correlate with the avoidance behavior. Instead, bloating of the intestinal lumen caused by the pathogen infection underlies the avoidance behavior via modulation of both DAF-7/TGF-β and the G-protein coupled receptor NPR-1 neuroendocrine pathways, which regulate aerotaxis. We further show that the modulation of these neuroendocrine pathways by intestinal infection or genetic modulation of aerotaxis by loss of *ocr-2* and *osm-9* drives the change in preference of animals from *P. aeruginosa* to *E. coli* lawns. Our findings demonstrate that signaling from the intestine via neuroendocrine pathways modulates microbial perception during infection.

## Results

### *P. aeruginosa* infection elicits *C. elegans* avoidance in a phenazine-independent manner

Because *P. aeruginosa*-produced phenazines lead to induction of the neuromodulator DAF-7/TGF-β in the ASJ neuron pair (*Meisel et al., 2014*), we studied the role of phenazines in the elicitation of the pathogen avoidance behavior. *P. aeruginosa* uses a well-characterized biosynthetic pathway to generate four different phenazines (*Dietrich et al., 2006*) (*Figure 1A*). Phenazine-1-carboxylic acid, the precursor of all other phenazines produced by *P. aeruginosa*, is synthesized from chorismate by

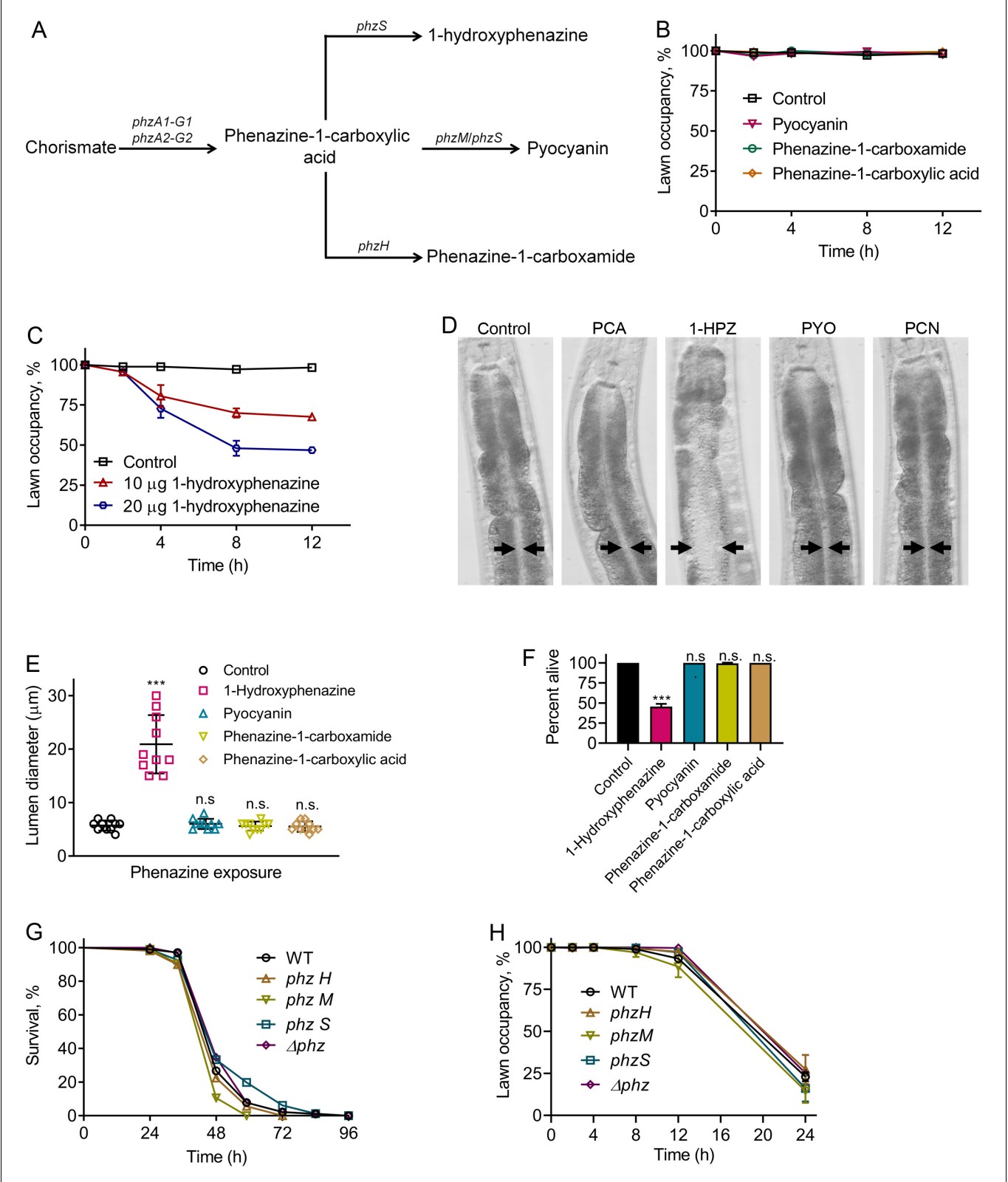

**Figure 1.** Phenazine-independent elicitation of *C. elegans* avoidance of *P. aeruginosa* during infection. (**A**) Phenazine synthesis pathway of *P. aeruginosa*. (**B**) Time course of the percent occupancy of N2 animals on *E. coli* lawns containing 20 μg of pyocyanin, phenazine-1-carboxamide, and phenazine-1-carboxylic acid. For the control, the animals were exposed to solvent mock *E. coli* lawns. (**C**) Time course of the percent occupancy of N2 animals on *E. coli* lawns containing 1-hydroxyphenazine. For the control, the animals were exposed to solvent mock *E. coli* lawns. (**D**) Representative

*Figure 1 continued on next page*

*Figure 1 continued*

photomicrographs of N2 animals exposed for 8 hr to *E. coli* lawns containing 20 µg of phenazine-1-carboxylic acid (PCA), 1-hydroxyphenazine (1-HPZ), pyocyanin (PYO), and phenazine-1-carboxamide (PCN). For the control, the animals were exposed for 8 hr to solvent mock *E. coli* lawns. Arrows point to the border of the intestinal lumen. (E) Quantification of the diameter of the intestinal lumen of N2 animals exposed for 8 hr to *E. coli* lawns containing 20 µg of different phenazines. ***p<0.001 via the t test. n.s., non-significant. (F) Percent of animals alive after 24 hr of exposure to *E. coli* lawns containing 20 µg of different phenazines. The bars show the means ± SD from three independent experiments. ***p<0.001 via the t test. n.s., non-significant. (G) Representative survival plots of N2 animals on different phenazine synthesis pathway mutants of *P. aeruginosa*. p-value relative to WT, n.s., non-significant. (H) Time course of the percent occupancy of N2 animals on lawns of different phenazine synthesis pathway mutants of *P. aeruginosa*.

a full set of functional phenazine-1-carboxylic acid biosynthetic enzymes encoded by the *phzA1-G1* and *phzA2-G2* operons (*Dietrich et al., 2006*). Phenazine-1-carboxylic acid can be modified by other enzymes to make 1-hydroxyphenazine, phenazine-1-carboxamide, or pyocyanin (*Figure 1A*). Phenazine-1-carboxamide, but not other phenazines, activates DAF-7/TGF-β expression in ASJ neurons (*Meisel et al., 2014*). We tested whether any of the four purified phenazines produced by *P. aeruginosa* elicited avoidance when added to lawns of *E. coli*. We found that animals exposed to *E. coli* lawns containing phenazine-1-carboxamide, which is required for the activation of DAF-7/TGF-β expression in ASJ neurons (*Meisel et al., 2014*), did not exhibit the avoidance behavior (*Figure 1B*). Additionally, the animals did not avoid *E. coli* lawns with added phenazine-1-carboxylic acid and pyocyanin (*Figure 1B*). In contrast, the animals did avoid lawns of *E. coli* containing 1-hydroxyphenazine (*Figure 1C*).

We observed that animals feeding on *E. coli* lawns that were supplemented with 1-hydroxyphenazine, but not other phenazines, showed bloating of the intestinal lumen (*Figure 1D,E*). Bloating of the intestine is known to elicit microbial avoidance behavior, including the avoidance of *E. coli* (*Kumar et al., 2019*; *Singh and Aballay, 2019a*). Therefore, the avoidance of *E. coli* lawns containing 1-hydroxyphenazine was likely caused by intestinal bloating of the animals induced by the toxin (*Figure 1D,E*). We also observed that animals feeding on *E. coli* lawns supplemented with 1-hydroxyphenazine, but not other phenazines, showed drastically reduced survival (*Figure 1F*). More than 50% of the animals feeding on *E. coli* lawns containing 1-hydroxyphenazine died within 24 hr, while animals feeding on control *E. coli* lawns or *E. coli* lawns containing other phenazines remained alive at the same time point (*Figure 1F*). However, *P. aeruginosa* mutants deficient in phenazine production were not compromised in their ability to kill *C. elegans* compared with wild-type *P. aeruginosa* (*Figure 1G*), suggesting that the addition of 1-hydroxyphenazine may have non-physiological effects. Moreover, all the phenazine mutants induced an avoidance behavior that was indistinguishable from that caused by the wild-type *P. aeruginosa* (*Figure 1H*). These results suggest that while high amounts of externally added 1-hydroxyphenazine elicit microbial avoidance behavior, the normal amount of phenazines produced by *P. aeruginosa* during infection is insufficient for induction of the avoidance behavior.

## *P. aeruginosa*-induced *daf-7* expression in ASJ neurons is insufficient for eliciting pathogen avoidance

*C. elegans* is initially attracted towards lawns of *P. aeruginosa*, and after an initial phase of interaction, the animals begin to avoid the bacterial lawns. However, the initial phase of interaction before elicitation of the avoidance behavior is variable, leading to different avoidance kinetics in various studies (*Hao et al., 2018*; *Hilbert and Kim, 2017*; *Ma et al., 2017*; *Martin et al., 2017*; *Meisel et al., 2014*; *Singh and Aballay, 2019a*; *Sun et al., 2011*). It is likely that the variations in avoidance kinetics are due to differences in the production of *P. aeruginosa* factors governing the avoidance behavior. We reasoned that different growing conditions for the *P. aeruginosa* lawns might be responsible for the differences in avoidance behaviors. We observed that one of the major differences in bacterial lawn preparations is the variation in incubation periods of *P. aeruginosa* on agar plates before animal exposure (*Hao et al., 2018*; *Hilbert and Kim, 2017*; *Ma et al., 2017*; *Meisel et al., 2014*; *Singh and Aballay, 2019a*; *Sun et al., 2011*). To examine whether the differences in culture conditions of *P. aeruginosa* on agar plates before transferring *C. elegans* could be the underlying reason for the different avoidance kinetics, we varied the *P. aeruginosa* incubation times (*Figure 2A*). The avoidance behavior of the animals was enhanced with the incubation period of the *P. aeruginosa* lawns (*Figure 2B*). Animals exposed to *P. aeruginosa* lawns that were incubated at 37°

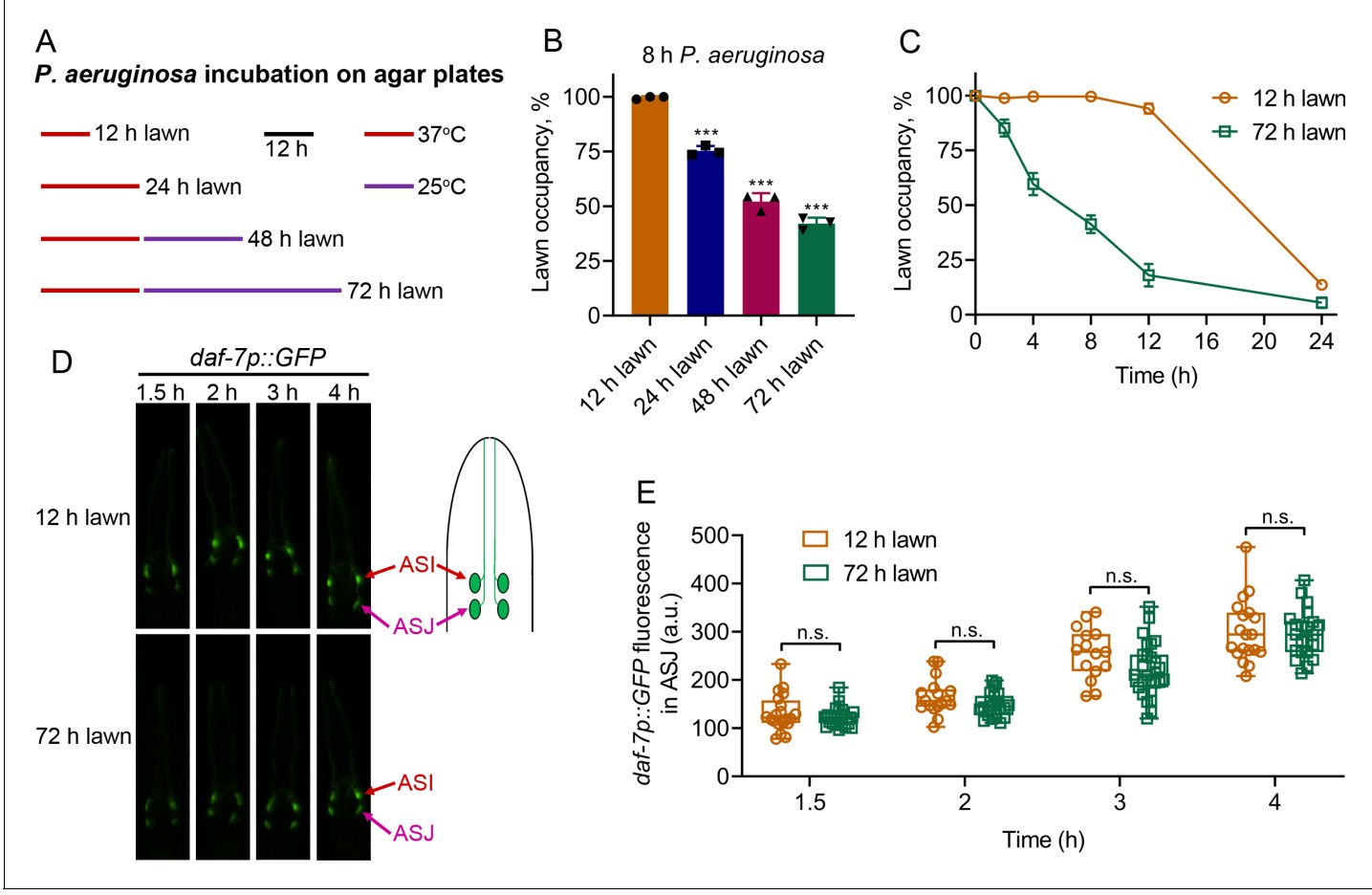

**Figure 2.** *P. aeruginosa*-induced *daf-7* expression in ASJ neurons is insufficient to elicit avoidance behavior. (A) Scheme for obtaining *P. aeruginosa* preparations with varying times and temperatures of incubation on SK plates. (B) Percent lawn occupancy of N2 animals after 8 hr of incubation on different preparations of *P. aeruginosa*. The black symbols represent individual data points. The bars show the means ± SD from three independent experiments. ***p<0.001 via the t test. (C) Time course of the percent occupancy of N2 animals on 12 and 72 hr lawns of *P. aeruginosa*. (D) Time course of induction of *daf-7p::GFP* on 12 and 72 hr lawns of *P. aeruginosa*. The ASI and ASJ chemosensory neurons are labeled. The drawing depicts the arrangement of the ASI and ASJ neurons in *C. elegans* head. (E) Quantification of induction of *daf-7p::GFP* in the ASJ chemosensory neuron on 12 and 72 hr lawns of *P. aeruginosa* over time. n.s., non-significant via the t test.

The online version of this article includes the following figure supplement(s) for figure 2:

**Figure supplement 1.** *P. aeruginosa*-induced *daf-7* expression in ASI neurons is indistinguishable on 12 and 72 hr lawns.

C for 24 hr followed by 25°C for 48 hr (referred to as 72 hr lawn) showed a significantly enhanced avoidance rate in comparison to animals exposed to *P. aeruginosa* lawns that were incubated at 37°C for 12 hr (referred to as 12 hr lawn) (*Figure 2C*).

We tested whether the differences in avoidance kinetics exhibited by the animals on 12 and 72 hr lawns could be due to differences in the induction of *daf-7* expression in ASJ neurons. We found that while animals exposed to 12 hr lawns showed robust induction of *daf-7* expression in ASJ neurons that did not differ from animals exposed to 72 hr lawns (*Figure 2D,E*), animals exposed to 12 hr lawns exhibited a delayed avoidance compared with animals exposed to 72 hr lawns (*Figure 2C*). In addition, the induction of *daf-7* expression in ASI neurons was not significantly different in animals exposed to 12 or 72 hr lawns (*Figure 2—figure supplement 1*). Because the induction of *daf-7* expression in ASJ neurons was indistinguishable in animals on 12 and 72 hr lawns, the results suggest that the induction of *daf-7* expression in ASJ may not be the only cause of the elicitation of avoidance behavior.

## Bloating of the intestinal lumen underlies the avoidance behavior towards *P. aeruginosa*

Because 1-hydroxyphenazine induces both avoidance behavior and intestinal bloating (*Figure 1C,D*), we tested whether bloating could account for the faster avoidance exhibited by animals exposed to the 72 hr lawn compared with those exposed to the 12 hr lawn. We found that animals exposed to 72 hr lawns showed bloated intestines as early as 8 hr, while the lumens of animals exposed to 12 hr lawns for 8 hr were comparable to those of animals fed *E. coli* (*Figure 3A–C*). Because bloating of the intestine leads to the induction of genes that are part of the NPR-1 neuroendocrine pathway (*Singh and Aballay, 2019a*), we examined the expression levels of the *npr-1*, *flp-18*, and *flp-21* genes. As shown in *Figure 3D*, animals exposed to 72 hr lawns, but not to 12 hr lawns, showed higher expression levels of the *npr-1*, *flp-18*, and *flp-21* genes compared with the control animals on *E. coli*.

To further confirm the relationship between avoidance behavior and intestinal bloating of animals exposed to 72 hr lawns, we studied avoidance in animals deficient in the *nol-6* gene. Previous studies have shown that RNA interference (RNAi)-mediated knockdown of *nol-6*, a nucleolar RNA-associated protein, reduces bloating of the intestinal lumen caused by bacterial infection (*Fuhrman et al., 2009*). We found that *nol-6* RNAi delayed pathogen avoidance (*Figure 3E*, *Figure 3—figure supplement 1A*). Animals deficient in *nol-6*, which failed to avoid *P. aeruginosa* at 8 hr (*Figure 3E*), did not exhibit intestinal bloating at the same time point when exposed to 72 hr lawns (*Figure 3F,G*). Consistent with the idea that bloating induces pathogen avoidance, *nol-6* RNAi animals exposed to 72 hr lawns avoided *P. aeruginosa* at 24 hr (*Figure 3E*), at which time they also exhibited intestinal bloating (*Figure 3—figure supplement 1B,C*). These results indicate that *nol-6* RNAi animals are not generally defective in avoidance behavior, and the delayed avoidance behavior is due to delayed intestinal bloating. Despite diminishing the avoidance behavior, knockdown of *nol-6* did not affect the induction of *daf-7* expression in either ASJ (*Figure 3H*) or ASI neurons (*Figure 3—figure supplement 1D*). Taken together, these results suggest that bloating of the intestine, but not induction of *daf-7* in ASJ neurons, underlies the avoidance behavior.

## *P. aeruginosa* virulence correlates with pathogen avoidance behavior

We next tested whether the enhanced intestinal bloating of animals exposed for 8 hr to 72 hr lawns was due to increased bacterial colonization compared with animals exposed to 12 hr lawns for the same time. We found that while animals exposed to 12 hr lawns showed a consistent increase in intestinal colonization, animals exposed to 72 hr lawns did not show any increase in colonization during the first 8 hr of exposure (*Figure 4A*). These results indicate that intestinal bloating on 72 hr lawns is independent of bacterial colonization. Thus, we investigated whether the survival of animals on the two types of lawns was different. Animals exposed to 72 hr lawns died significantly faster than animals exposed to 12 hr lawns (*Figure 4B*), indicating that the virulence of *P. aeruginosa* was higher on 72 hr than on 12 hr lawns.

It has been shown that a non-virulent *P. aeruginosa* strain deficient in the gene *gacA*, a global activator of gene expression and virulence, fails to elicit *C. elegans* avoidance (*Hao et al., 2018*; *Singh and Aballay, 2019a*). *P. aeruginosa gacA* mutants are also hampered in the induction of *daf-7* in the ASJ chemosensory neuron (*Meisel et al., 2014*). Thus, it is not clear whether the inability of *P. aeruginosa gacA* mutants to elicit an avoidance behavior is due to reduced *daf-7* induction in the ASJ neuron or to their reduced virulence. To distinguish between these two possibilities and to test the role of *P. aeruginosa* virulence in *C. elegans* avoidance behavior, we tested the induction of *daf-7* in ASJ neurons and avoidance behavior elicited by several strains of *P. aeruginosa* with reduced virulence. We selected a diverse set of *P. aeruginosa* mutants that have attenuated virulence in *C. elegans* (*Feinbaum et al., 2012*). We confirmed that all the studied *P. aeruginosa* mutants exhibited reduced virulence compared with wild-type *P. aeruginosa* (*Figure 4—figure supplement 1A*). All of these mutants were also deficient in the elicitation of pathogen avoidance behavior (*Figure 4C*). We next tested the induction of *daf-7::GFP* in the ASJ neurons upon exposure to the aforementioned *P. aeruginosa* mutants. We found that while *kinB* and *rhlR* mutants of *P. aeruginosa* were deficient in the induction of *daf-7* in ASJ neurons, the induction of *daf-7* by *lasR*, *lysC,* and *ptsP* mutants was comparable to the induction by wild-type *P. aeruginosa* (*Figure 4D,E*). The induction of *daf-7* expression in ASI neurons by all of these *P. aeruginosa* mutants was indistinguishable from the

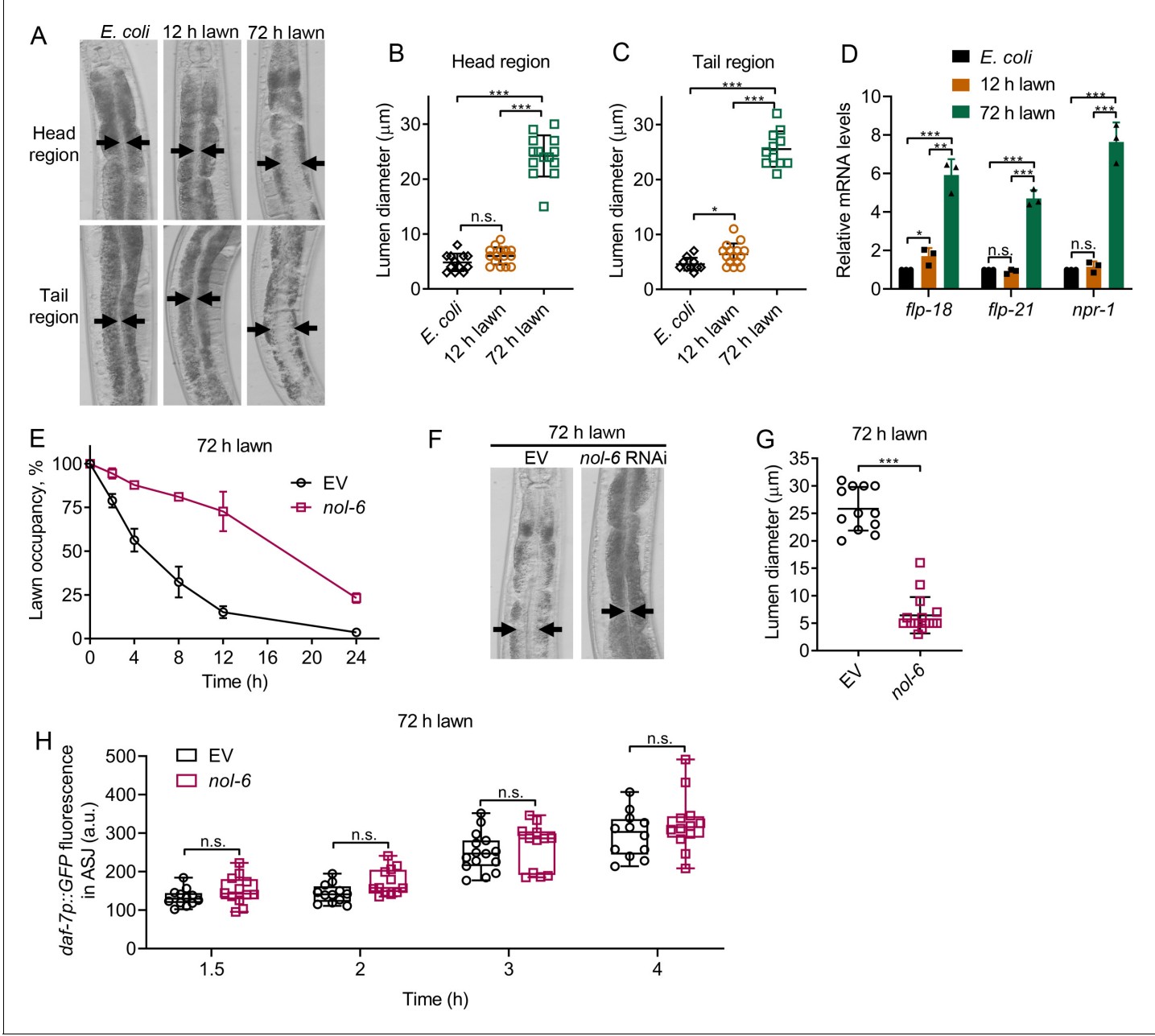

**Figure 3.** Intestinal lumen bloating underlies the avoidance behavior towards *P. aeruginosa*. (**A**) Representative photomicrographs of N2 animals exposed for 8 hr to *E. coli* lawns, and 12 and 72 hr lawns of *P. aeruginosa*. Representative photomicrographs of the head and tail regions are shown. Arrows point to the border of the intestinal lumen. (**B and C**) Quantification of the diameter of the intestinal lumen of N2 animals exposed for 8 hr to *E. coli* lawns, and 12 and 72 hr lawns of *P. aeruginosa* from the head (**B**) and tail (**C**) regions. ***p<0.001 and *p<0.05 via the t test. n.s., non-significant. (**D**) Gene expression analysis of N2 animals grown on *E. coli* until the young adult stage, followed by incubation for 8 hr on *E. coli* lawns, and 12 and 72 hr lawns of *P. aeruginosa*. The black symbols represent individual data points. ***p<0.001, **p<0.01, and *p<0.05 via the t test. n.s., non-significant. (**E**) Time course of the percent occupancy of the control (EV) as well as *nol-6* RNAi animals on 72 hr lawns of *P. aeruginosa*. (**F**) Representative photomicrographs of N2 animals grown on control and *nol-6* RNAi exposed for 8 hr to 72 hr lawns of *P. aeruginosa*. Arrows point to the border of the intestinal lumen. (**G**) Quantification of the diameter of the intestinal lumen of N2 animals grown on control and *nol-6* RNAi exposed for 8 hr to 72 hr lawns of *P. aeruginosa*. ***p<0.001 via the t test. (**H**) Time course of induction of *daf-7p::GFP* in ASJ neurons in animals grown on control and *nol-6* RNAi and exposed to 72 hr lawns of *P. aeruginosa*. n.s., non-significant via the t test.

The online version of this article includes the following figure supplement(s) for figure 3:

**Figure supplement 1.** Intestinal lumen bloating underlies the avoidance behavior towards *P. aeruginosa*.

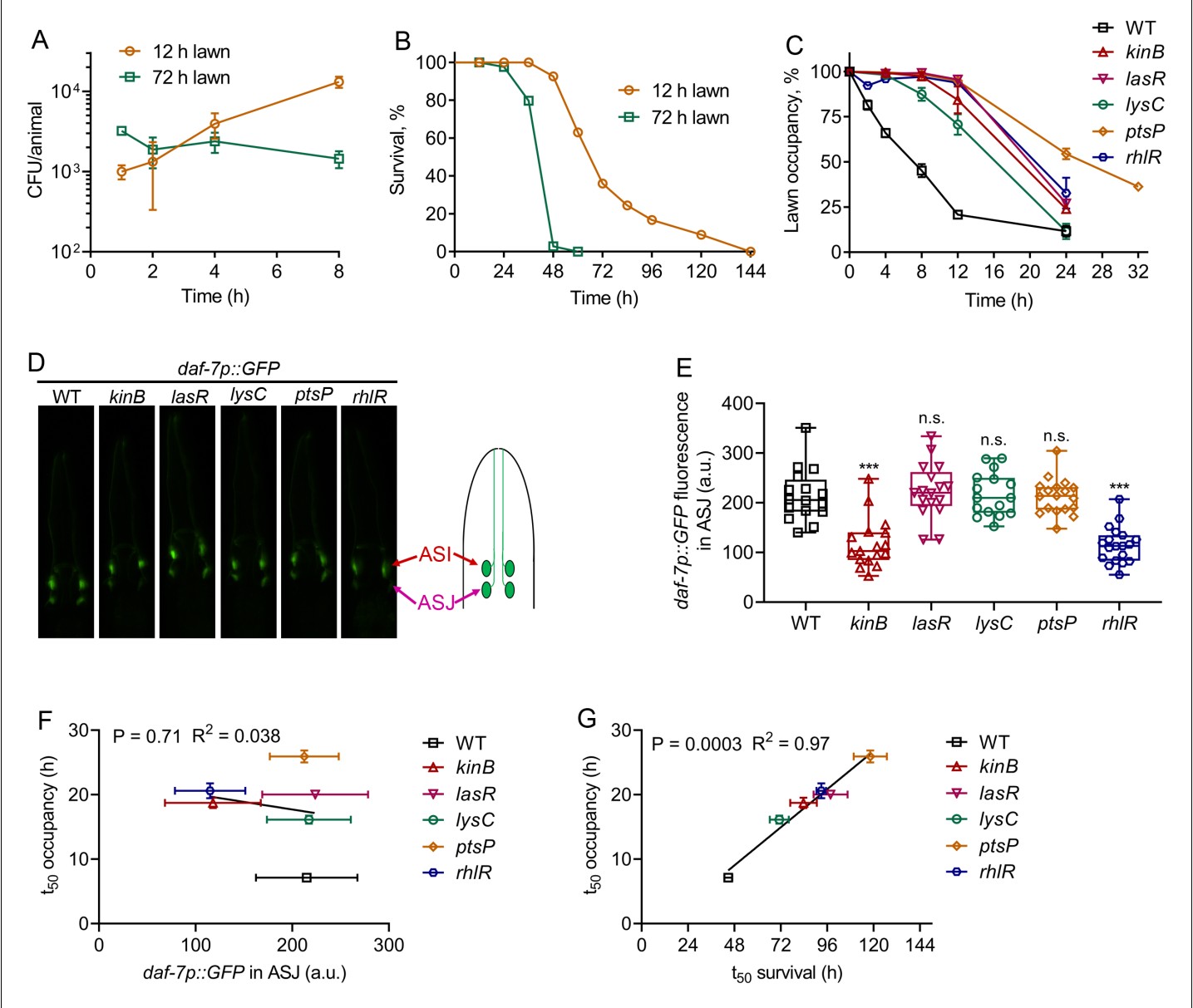

**Figure 4.** *P. aeruginosa* virulence correlates with the avoidance behavior. (**A**) Time course of colony-forming units (CFU) per animal of N2 animals exposed to 12 and 72 hr lawns of *P. aeruginosa*-GFP. (**B**) Representative survival plots of N2 animals on 12 and 72 hr lawns of *P. aeruginosa*. p<0.0001. (**C**) Time course of the percent occupancy of N2 animals on 72 hr lawns of different mutants of *P. aeruginosa*. (**D**) Representative photomicrographs of *daf-7p::GFP* expressing animals exposed for 4 hr to lawns of different mutants of *P. aeruginosa*. The drawing depicts the arrangement of the ASI and ASJ neurons in *C. elegans* head. (**E**) Quantification of *daf-7p::GFP* in the ASJ chemosensory neuron pair in animals exposed for 4 hr to lawns of different mutants of *P. aeruginosa*. ***p<0.001 via the t test. n.s., non-significant. (**F**) Correlation of the mean lawn occupancy time (t$_{50}$ occupancy) to the corresponding levels of *daf-7p::GFP* in the ASJ chemosensory neuron pair in animals exposed to different *P. aeruginosa* mutants. (**G**) Correlation of the mean lawn occupancy time (t$_{50}$ occupancy) to the corresponding mean survival time (t$_{50}$ survival) in animals exposed to different *P. aeruginosa* mutants. The online version of this article includes the following figure supplement(s) for figure 4:

**Figure supplement 1.** *P. aeruginosa* mutants with reduced virulence induce *daf-7p::GFP* in ASI neurons.

induction by wild-type *P. aeruginosa* (**Figure 4—figure supplement 1B**). The levels of *daf-7* induction in the ASJ neurons induced by different *P. aeruginosa* mutants did not show any correlation with the mean occupancy of the animals on *P. aeruginosa* mutant lawns (**Figure 4F**). In contrast, the mean survival of the animals on different *P. aeruginosa* mutants showed a strong correlation with the mean occupancy of the animals on *P. aeruginosa* mutant lawns (**Figure 4G**). Taken together, these

results show that the virulence of *P. aeruginosa*, and not its ability to induce *daf-7* in ASJ neurons, correlates with *C. elegans* avoidance behavior.

## Neuroendocrine signaling involved in the control of aerotaxis behavior regulates associative learning of pathogens

Based on the finding that intestinal bloating caused by infection, and not chemosensation of *P. aeruginosa* phenazines, elicits the avoidance behavior, we hypothesized that intestinal infection may be responsible for the associative learning of pathogens. It is known that prior exposure to *P. aeruginosa* for several hours enables *C. elegans* to preferentially choose nonpathogenic *E. coli* over *P. aeruginosa* (*Zhang et al., 2005*). Similarly, in a two-choice assay, when naïve animals were given a choice between *E. coli* and *P. aeruginosa*, the animals changed their preference from *P. aeruginosa* to *E. coli* after 8 hr of exposure (*Figure 5A,B*). The *P. aeruginosa* choice index (CI), described in *Figure 5A*, measures the preference of animals for *P. aeruginosa* with values ranging from −1 to 1. The values 1,–1, and 0 indicate that all animals are on *P. aeruginosa*, all animals are away from *P. aeruginosa*, and an equal number of animals is on *P. aeruginosa* and *E. coli*, respectively. Because aerotaxis plays a role in pathogen avoidance (*Meisel et al., 2014*; *Reddy et al., 2009*; *Singh and Aballay, 2019a*; *Styer et al., 2008*), we examined whether aerotaxis is also important for changes in preference from pathogenic to nonpathogenic bacteria.

We reasoned that if animals experience different levels of oxygen on lawns of different bacteria, aerotaxis-regulating pathways might affect the microbial preference. It is known that *P. aeruginosa* lawns have lower oxygen levels than *E. coli* lawns (*Gray et al., 2004*; *Reddy et al., 2011*). To determine whether the animals on *E. coli* and *P. aeruginosa* respond to the different levels of oxygen of the two types of lawns, we used the *cysl-2p::GFP* reporter strain. Expression of the gene *cysl-2* is regulated by hypoxia-inducible factor 1 (HIF-1), a transcription factor that is induced by low levels of oxygen (*Ma et al., 2012*). Since HIF-1 is degraded at higher and accumulates at lower oxygen levels (*Jiang et al., 2001*), the expression levels of *cysl-2* correlate inversely with the oxygen levels experienced by the animals. We found that animals exposed for 24 hr to *P. aeruginosa* lawns had higher GFP levels compared with those exposed to *E. coli* lawns (*Figure 5C,D*).

We observed that the loss of function mutants *daf-7* and *npr-1*, which are deficient in DAF-7/TGF-β, and NPR-1 signaling, respectively, inhibited both pathogen avoidance (*Figure 5—figure supplement 1A,B*) and the change in preference from *P. aeruginosa* to *E. coli* (*Figure 5E,F*). These two neuroendocrine pathways are known to act in parallel (*Chang et al., 2006*; *de Bono et al., 2002*); therefore, we investigated the behaviors of *daf-7(ok3125);npr-1(ad609)* animals. We found that the aforementioned phenotypes were enhanced in *daf-7(ok3125);npr-1(ad609)* animals (*Figure 5G*, *Figure 5—figure supplement 1C*). Taken together, these results suggest that aerotaxis-regulating pathways are required for the change in microbial preference upon pathogen infection.

The genetic interactions for oxygen preference have been well characterized in *C. elegans* (*Chang et al., 2006*; *Chang and Bargmann, 2008*). The increased preference for low oxygen in loss of function *npr-1* and *daf-7* mutants requires the function of the transient receptor potential channel vanilloid (TRPV) genes *osm-9* and *ocr-2* (*Chang et al., 2006*). Because loss of function *ocr-2* and *osm-9* mutants have an increased preference for high oxygen levels (*Chang et al., 2006*), and because *E. coli* lawns have relatively higher oxygen levels (*Gray et al., 2004*; *Reddy et al., 2011*), we reasoned that these mutants should rapidly change their preference to *E. coli* if given the choice between *E. coli* and *P. aeruginosa*. First, we studied whether the loss of function mutants *ocr-2(ak47)* and *osm-9(yz6)* showed enhanced avoidance of *P. aeruginosa* lawns. As expected, *ocr-2(ak47)* and *osm-9(yz6)* animals exhibited a strong enhancement of avoidance behavior (*Figure 6A*, *Figure 6—figure supplement 1A*). These animals also showed a rapid change in preference to *E. coli* lawns in the two-choice assay (*Figure 6B*, *Figure 6—figure supplement 1B*). The preference for high oxygen in *ocr-2(ak47)* and *osm-9(yz6)* mutants is suppressed by the loss of function mutation in *egl-9*, a negative regulator of HIF-1 (*Chang and Bargmann, 2008*). Consistent with the function of EGL-9, we observed that the *egl-9* mutation suppressed both the enhanced pathogen avoidance and *E. coli* preference of *osm-9(yz6)* animals (*Figure 6C,D*). These results show that aerotaxis regulates both *P. aeruginosa* avoidance and changes in microbial preference.

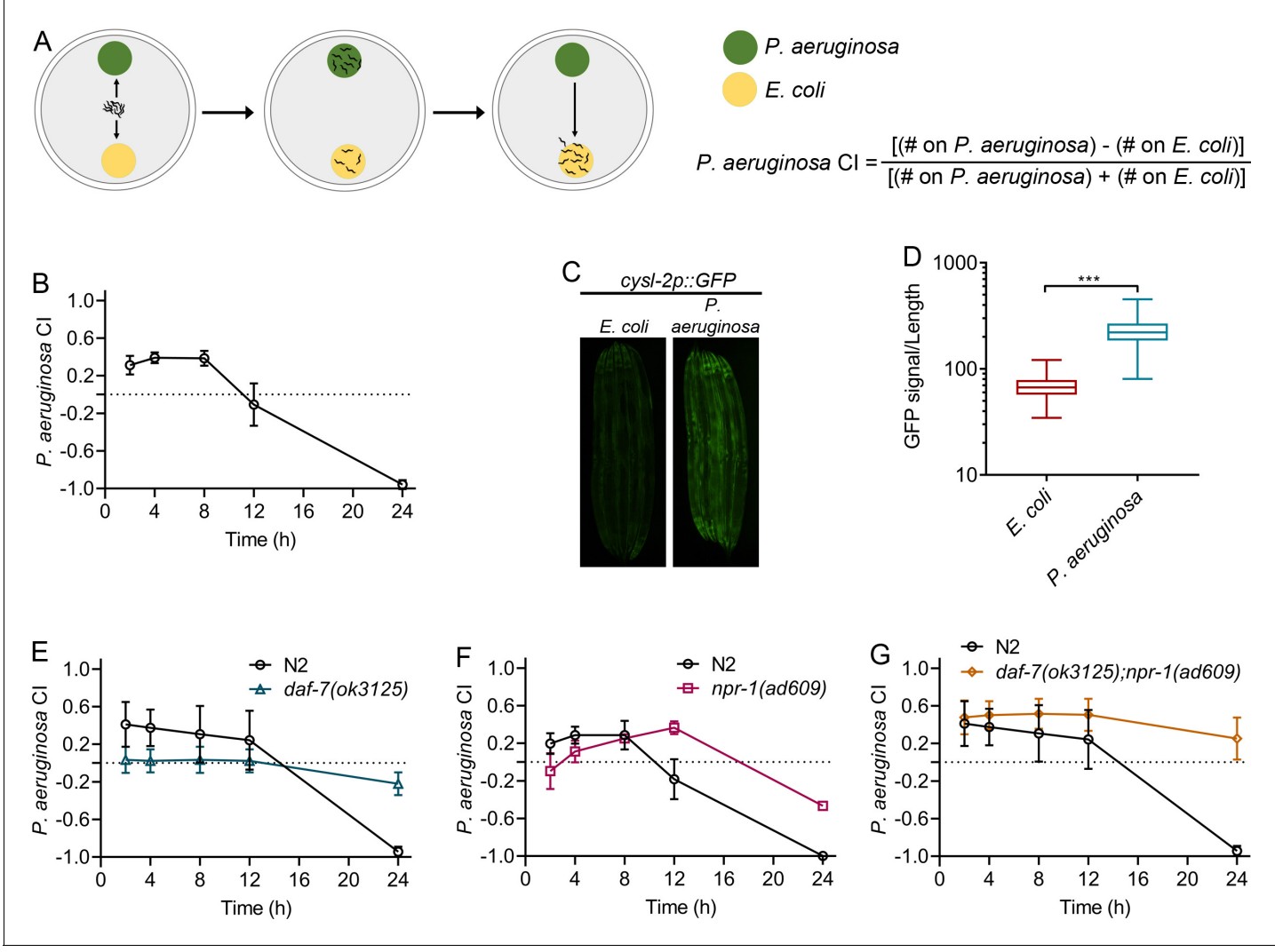

**Figure 5.** Neuroendocrine signaling involved in the control of aerotaxis behavior regulates associative learning of pathogens. (A) Schematic representation of the two-choice preference assay. Animals are transferred to the center of plates equidistant from the lawns of *P. aeruginosa* and *E. coli*. The number of animals on both lawns is counted at a given time and used to calculate the *P. aeruginosa* choice index (CI). (B) Time course of the *P. aeruginosa* CI of N2 animals in a two-choice preference assay containing one lawn of each *P. aeruginosa* and *E. coli*. (C) Representative photomicrographs of *cysl-2p::GFP*-expressing animals exposed for 24 hr to *E. coli* and *P. aeruginosa* lawns. (D) Quantification of *cysl-2p::GFP* levels in animals exposed for 24 hr to *E. coli* and *P. aeruginosa* lawns. The quantification was conducted using a COPAS Biosort machine to measure the mean GFP signal and the length of individual animals. The GFP signal of each animal was normalized to its length. The data are plotted as a box and whisker plot from over 100 animals for each condition. ***p<0.001 via the t test. (E) Time course of the *P. aeruginosa* CI of N2 and *daf-7(ok3125)* animals in a two-choice preference assay containing one lawn of each *P. aeruginosa* and *E. coli*. (F) Time course of the *P. aeruginosa* CI of N2 and *npr-1(ad609)* animals in a two-choice preference assay containing one lawn of each *P. aeruginosa* and *E. coli*. (G) Time course of the *P. aeruginosa* CI of N2 and *daf-7 (ok3125);npr-1(ad609)* animals in a two-choice preference assay containing one lawn of each *P. aeruginosa* and *E. coli*.

The online version of this article includes the following figure supplement(s) for figure 5:

**Figure supplement 1.** Aerotaxis behavior through neuroendocrine signaling controls microbial avoidance behavior.

## Intestinal bloating controls the change in microbial preference upon infection by activating aerotaxis pathways

While the above results showed that aversive learning and changes in microbial preference require aerotaxis pathways, they do not provide insights into the signaling during microbial infection that leads to changes in microbial preference. Bloating of the intestinal lumen upon pathogen infection activates the NPR-1/GPCR and DAF-7/TGF-β pathways, which results in a preference towards high oxygen (*Singh and Aballay, 2019a*). Thus, we reasoned that animals with bloated intestines should

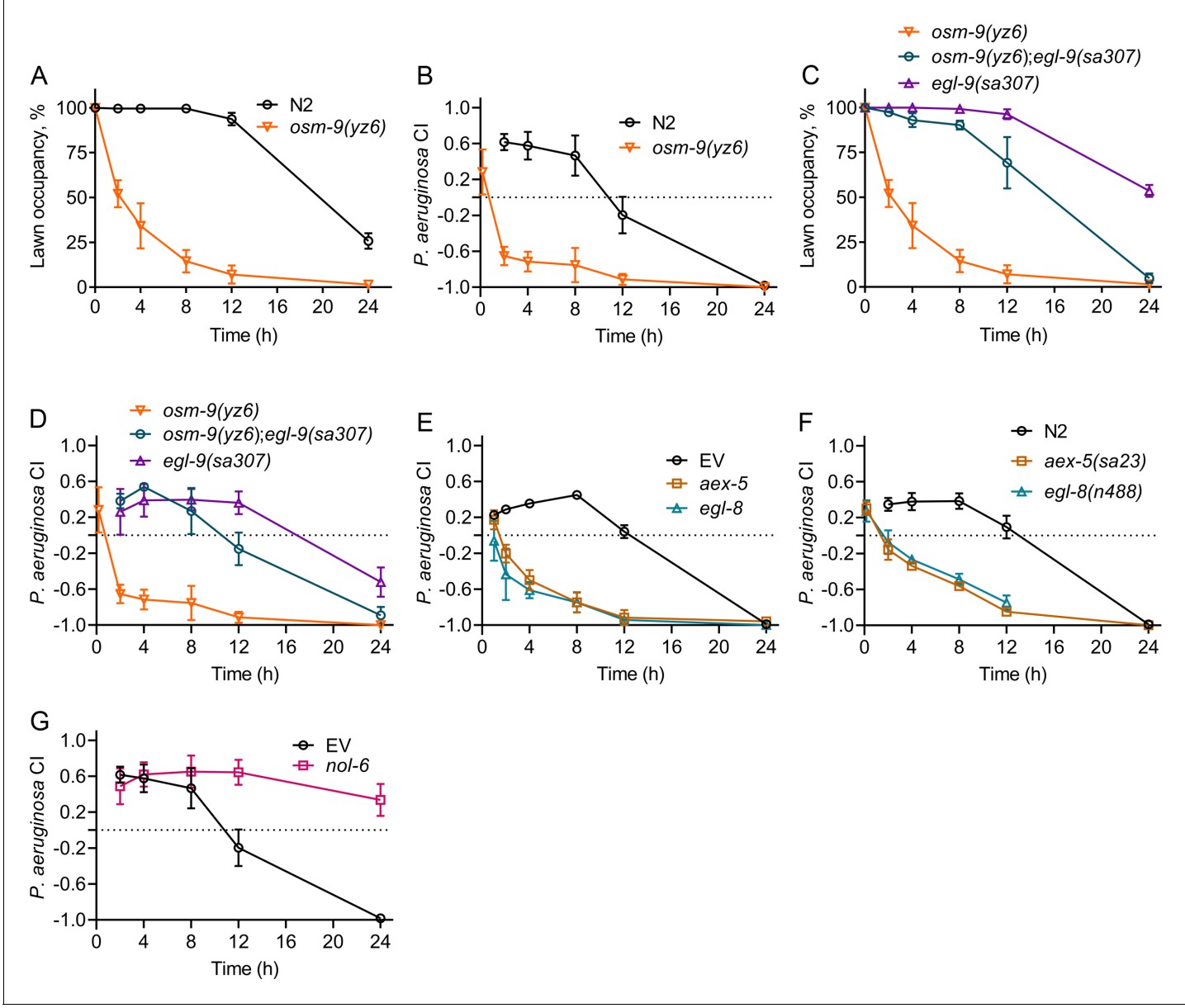

**Figure 6.** Modulation of aerotaxis behavior by intestinal bloating underlies the change in microbial preference upon infection. (A) Time course of the percent occupancy of N2 and *osm-9(yz6)* animals on *P. aeruginosa* lawns. (B) Time course of the *P. aeruginosa* CI of N2 and *osm-9(yz6)* animals in a two-choice preference assay containing one lawn of each *P. aeruginosa* and *E. coli*. (C) Time course of the percent occupancy of *osm-9(yz6)*, *osm-9(yz6);egl-9(sa307)*, and *egl-9(sa307)* animals on *P. aeruginosa* lawns. (D) Time course of the *P. aeruginosa* CI of *osm-9(yz6)*, *osm-9(yz6);egl-9(sa307)*, and *egl-9 (sa307)* animals in a two-choice preference assay containing one lawn of each *P. aeruginosa* and *E. coli*. (E) Time course of the *P. aeruginosa* CI of N2 animals grown on RNAi control bacteria, as well as bacteria for RNAi against *aex-5* and *egl-8* in a two-choice preference assay containing one lawn of each *P. aeruginosa* and *E. coli*. EV, empty vector RNAi control. (F) Time course of the *P. aeruginosa* CI of N2, *aex-5(sa23)*, and *egl-8(n488)* animals in a two-choice preference assay containing one lawn of each *P. aeruginosa* and *E. coli*. (G) Time course of the *P. aeruginosa* CI of N2 animals grown on *nol-6* RNAi as well as control bacteria in a two-choice preference assay containing one lawn of each *P. aeruginosa* and *E. coli*. EV, empty vector RNAi control.

The online version of this article includes the following figure supplement(s) for figure 6:

**Figure supplement 1.** Modulation of aerotaxis behavior alters microbial preference.

**Figure supplement 2.** Modulation of aerotaxis alters microbial preference.

**Figure supplement 3.** Low oxygen levels do not affect intestinal colonization and bloating.

show much more rapid learning and change in preference to *E. coli* in the two-choice assay. We examined the change in preference from *P. aeruginosa* to *E. coli* in the two-choice assay of *aex-5* and *egl-8* knockdown animals. Knockdown of these genes caused bloating of the intestinal lumen and led to enhanced avoidance of *P. aeruginosa* (*Singh and Aballay, 2019a*). As shown in *Figure 6E,F*, inhibition of these genes by RNAi and mutations also elicited a rapid change in preference to *E. coli* lawns in the two-choice assay. We were also able to elicit a rapid preference towards *E. coli* by exposing the animals to 5% oxygen (*Figure 6—figure supplement 2*). Exposure to low oxygen levels alone does not affect either intestinal bacterial colonization or bloating (*Figure 6—figure supplement 3*).

Because bloating accelerates the preference of the animals towards *E. coli,* we predicted that animals resistant to infection and bloating would not be capable of changing bacterial preferences. We studied the change in microbial preference in *nol-6* RNAi animals that are resistant to *P. aeruginosa* infection and bloating. As shown in *Figure 6G*, *nol-6* RNAi animals were defective in aversive learning toward *P. aeruginosa*. Together, these results suggest that intestinal bloating caused by pathogen infection, which modulates aerotaxis-regulating neuroendocrine pathways, is important for the learning process that leads to change in microbial preference from bacterial lawns containing relatively lower to those containing relatively higher oxygen levels.

## Discussion

Our study establishes that intestinal infection and bloating of the lumen, which depend on the virulence of *P. aeruginosa*, regulate both pathogen avoidance and aversive learning by modulating the neuroendocrine pathways NPR-1/GPCR and DAF-7/TGF-β that control aerotaxis behavior (*Figure 7*). Enhanced activities of these pathways, as a consequence of intestinal infection, lead to the avoidance of low oxygen, resulting in the avoidance of bacterial lawns with low oxygen due to microbial metabolism. Intestinal infection-mediated avoidance of low oxygen also drives the movement of animals from *P. aeruginosa* to *E. coli* lawns, which have been reported to have relatively higher oxygen

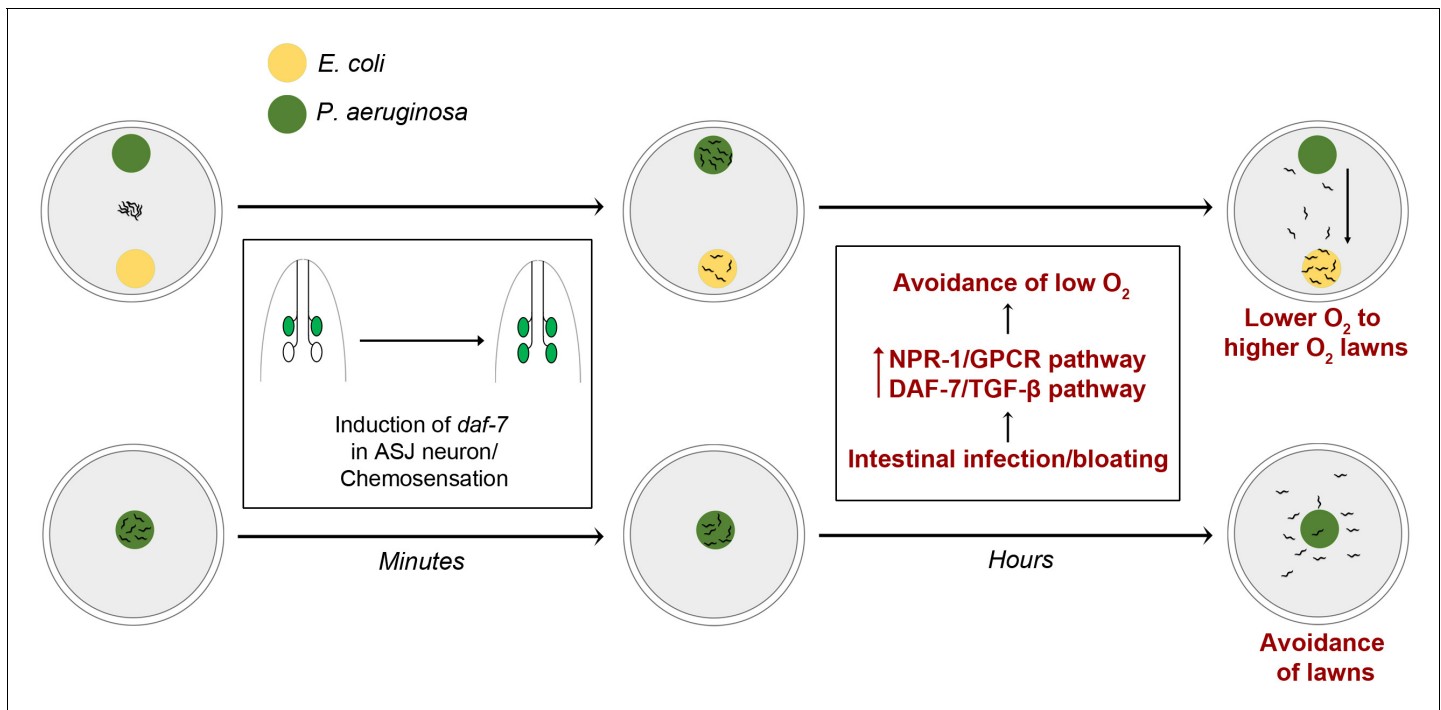

**Figure 7.** Model for intestinal infection-regulated microbial perception. Rapid chemosensation of *P. aeruginosa* resulting in the induction of *daf-7* expression in the ASJ neuron pair is insufficient for eliciting avoidance behavior. Intestinal infection leading to bloating activates NPR-1/GPCR and DAF-7/TGF-β neuroendocrine pathways, driving the evacuation of low O₂ *P. aeruginosa* lawns and change in preference from relatively lower O₂ lawns of *P. aeruginosa* to relatively higher O₂ lawns of *E. coli*.

levels. Thus, modulation of aerotaxis-regulating neuroendocrine pathways by intestinal infection plays a role in the learning process, resulting in changes in the preference of animals from *P. aeruginosa* to *E. coli* (*Figure 7*). Our study also indicates that microbial perception is controlled by inputs from the intestine during infection in *C. elegans*.

*P. aeruginosa* secondary metabolites, including phenazine-1-carboxamide, have been shown to activate the DAF-7/TGF-β pathway in the chemosensory neuron pair ASJ (*Meisel et al., 2014*). However, we found that phenazine-1-carboxamide does not play a role in elicitation of the avoidance behavior (*Figure 1B,H*). The induction of DAF-7/TGF-β in ASJ chemosensory neurons was observed within 6 min of exposure to *P. aeruginosa*, and no further changes were observed up to 24 hr (*Meisel et al., 2014*). However, the avoidance behavior was observed only after hours of interaction with *P. aeruginosa*. Therefore, while the activity of the DAF-7/TGF-β pathway is required to elicit the avoidance behavior (*Figure 5—figure supplement 1A*), its rapid activation in ASJ neurons by chemosensation appears to be insufficient for induction of this behavior. Because the DAF-7/TGF-β and NPR-1/GPCR pathways are induced by intestinal bloating (*Singh and Aballay, 2019a*) and act synergistically to elicit pathogen avoidance (*Figure 5G*, *Figure 5—figure supplement 1C*), it is likely that animals integrate multiple inputs, including intestinal infection and chemosensation, to induce defense responses. Indeed, in addition to $O_2$, animals sense $CO_2$ and NO from bacterial lawns via multiple chemosensory neurons to control the pathogen avoidance behavior (*Brandt and Ringstad, 2015*; *Hao et al., 2018*).

*C. elegans* possesses an innate attraction to the smell of *P. aeruginosa,* and a brief exposure to *P. aeruginosa* does not lead to changes in preference to *E. coli* (*Ha et al., 2010*; *Zhang et al., 2005*). Because the olfactory aversive learning of *P. aeruginosa* and changes in preference to *E. coli* require several hours of exposure to *P. aeruginosa*, the mechanism by which the animals learn to avoid pathogens over time remain unclear. Here we show that bloating of the intestine caused by *P. aeruginosa* infection is required for the learning process. We demonstrate that the learning involves modulation of aerotaxis-regulating neuroendocrine pathways by intestinal bloating. By genetic modulation of the aerotaxis behavior of the animals, we are able to induce a rapid change in preference from *P. aeruginosa* to *E. coli*.

Feeding on several pathogenic, but not nonpathogenic, bacteria leads to colonization and bloating of the intestine (*Fuhrman et al., 2009*; *Irazoqui et al., 2010*; *Kurz et al., 2003*; *Yuen and Ausubel, 2018*). Thus, the bloating-induced defense response might be a generalized response of *C. elegans* to infection by different pathogens and enables the animals to distinguish between pathogenic and innocuous microbes. Indeed, the induction of intestinal bloating by genetic manipulation activates avoidance behavior towards even nonpathogenic bacteria (*Kumar et al., 2019*; *Singh and Aballay, 2019a*). Interestingly, the neuropeptide Y (NPY)-related signaling neuroendocrine pathway NPR-1, which is activated by intestinal bloating in *C. elegans*, appears to have a conserved mode of activation and action across a variety of disparate species (*Duvall et al., 2019*; *Singh and Aballay, 2019b*). For instance, gut distension caused by a complete blood meal may activate the NPY-related signaling in mosquitoes and induce host aversion (*Singh and Aballay, 2019b*). It will be interesting to study whether intestinal bloating is a conserved danger signal that activates immune responses in different species.

The precise mechanism by which bacterial pathogens cause bloating remains unclear. *C. elegans* possesses a rhythmic defecation cycle that is timed by an oscillation in intestinal pH (*Pfeiffer et al., 2008*). Defects in the pH wave lead to defects in the defecation cycle causing intestinal bloating (*Pfeiffer et al., 2008*). Pathogenic bacteria such as *P. aeruginosa* and *Enterococcus faecalis*, but not nonpathogenic *E. coli*, disturb the pH wave in *C. elegans* intestine (*Benomar et al., 2018*). It is possible that it is this disturbance of pH wave by pathogens that leads to intestinal bloating in *C. elegans*. However, it remains to be determined how pathogenic bacteria disturb the pH wave. Replication of bacteria in the intestinal lumen alone and other possible mechanisms could also lead to intestinal bloating.

Increasing evidence shows that *C. elegans* recognizes potential pathogen attack by sensing homeostasis perturbations, including perturbations of core cellular processes, DNA damage, and intestinal bloating (*Dunbar et al., 2012*; *Ermolaeva et al., 2013*; *Kumar et al., 2019*; *McEwan et al., 2012*; *Melo and Ruvkun, 2012*; *Pukkila-Worley, 2016*; *Singh and Aballay, 2019a*). Thus, these studies show that the physiological changes induced by pathogen infection or toxins generate modifications in *C. elegans* homeostasis that elicit an innate immune response and

microbial aversion behavior via neuroendocrine signaling. It remains to be determined whether *C. elegans* is also capable of recognizing infecting microbes through more traditional mechanisms like those capable of recognizing microbe-associated molecular patterns (MAMPs). *C. elegans* mounts immune responses toward heat-killed pathogenic bacteria (*Pukkila-Worley and Ausubel, 2012*; *Yuen and Ausubel, 2018*), suggesting the possible existence of MAMP-like mechanisms. However, the interpretation of these results is not straightforward because heat-killed bacteria could lead to intestinal bloating and activation of defense responses (*Singh and Aballay, 2019a*).

Our studies indicate that chemosensation of *P. aeruginosa* phenazines, which leads to the rapid induction of DAF-7/TGF-β in ASJ chemosensory neurons in *C. elegans*, is insufficient for the elicitation of pathogen avoidance behavior. We cannot rule out the possibility that other metabolites or virulence factors play a role in the elicitation of pathogen avoidance. Our results suggest that intestinal bloating caused by microbial infection plays a crucial role in aversive learning and microbial preference from pathogenic *P. aeruginosa* to nonpathogenic *E. coli*. Recent studies in *Drosophila melanogaster* also highlight the role of intestinal infection in modulating the immune response and social behavior (*Chen et al., 2019*). Therefore, the modulation of behavior by intestinal infection appears to be conserved across species. Future research further deciphering these conserved pathways will aid in better understanding intestinal infection and gut dysbiosis-mediated behavioral and physiological changes.

## Materials and methods

**Key resources table**

| Reagent type (species) or resource | Designation | Source or reference | Identifiers | Additional information |
|---|---|---|---|---|
| Strain, strain background (*Escherichia coli*) | OP50 | *Caenorhabditis* Genetics Center (CGC) | OP50 | |
| Strain, strain background (*E. coli*) | HT115 | Source BioScience | HT115 | |
| Strain, strain background (*E. coli*) | DH5α-GFP | Frederick M Ausubel laboratory | DH5α-GFP | |
| Strain, strain background (*Pseudomonas aeruginosa*) | PA14 | Frederick M Ausubel laboratory | PA14 | |
| Strain, strain background (*P. aeruginosa*) | PA14-GFP | Frederick M Ausubel laboratory | PA14-GFP | |
| Strain, strain background (*P. aeruginosa*) | PA14 kinB | Deborah Hung laboratory | PA14 kinB | |
| Strain, strain background (*P. aeruginosa*) | PA14 lasR | Deborah Hung laboratory | PA14 lasR | |
| Strain, strain background (*P. aeruginosa*) | PA14 lysC | Jason A Papin laboratory | PA14 lysC | |
| Strain, strain background (*P. aeruginosa*) | PA14 rhlR | Thomas K Wood laboratory | PA14 rhlR | |
| Strain, strain background (*P. aeruginosa*) | PA14 phz | Lars Dietrich laboratory | PA14 phz | Lacks both phzA1-G1 and phzA2-G2 operons |
| Strain, strain background (*P. aeruginosa*) | PA14 phzH | Lars Dietrich laboratory | PA14 phzH | |

*Continued on next page*

*Continued*

| Reagent type (species) or resource | Designation | Source or reference | Identifiers | Additional information |
|---|---|---|---|---|
| Strain, strain background (*P. aeruginosa*) | PA14 phzM | Lars Dietrich laboratory | PA14 phzM | |
| Strain, strain background (*P. aeruginosa*) | PA14 phzS | Lars Dietrich laboratory | PA14 phzS | |
| Strain, strain background (*P. aeruginosa*) | PA14 ptsP | Meta Kuehn laboratory | PA14 ptsP | |
| Strain, strain background (*Caenorhabditis elegans*) | N2 Bristol | CGC | N2 | |
| Strain, strain background (*C. elegans*) | ksIs2 [daf-7p::GFP + rol-6(su1006)] | CGC | FK181 | |
| Strain, strain background (*C. elegans*) | nIs470 [cysl-2p::GFP + myo-2p::mCherry] | CGC | DMS640 | |
| Strain, strain background (*C. elegans*) | ocr-2(ak47) | CGC | CX4544 | |
| Strain, strain background (*C. elegans*) | osm-9(yz6) | CGC | JY190 | |
| Strain, strain background (*C. elegans*) | aex-5(sa23) | CGC | JT23 | |
| Strain, strain background (*C. elegans*) | egl-8(n488) | CGC | MT1083 | |
| Strain, strain background (*C. elegans*) | egl-9(sa307) | CGC | JT307 | |
| Strain, strain background (*C. elegans*) | npr-1(ad609) | CGC | DA609 | |
| Strain, strain background (*C. elegans*) | daf-7(ok3125) | CGC | RB2302 | |
| Strain, strain background (*C. elegans*) | osm-9(yz6);egl-9(sa307) | This study | | Materials and methods section |
| Strain, strain background (*C. elegans*) | daf-7(ok3125);npr-1(ad609) | This study | | Materials and methods section |
| Software, algorithm | GraphPad Prism 8 | GraphPad Software | RRID:SCR_002798 | https://www.graphpad.com/scientific software/prism/ |
| Software, algorithm | Photoshop CS5 | Adobe | RRID:SCR_014199 | https://www.adobe.com/products/photoshop.html |
| Software, algorithm | ImageJ | NIH | RRID:SCR_003070 | https://imagej.nih.gov/ij/ |
| Software, algorithm | Leica LAS v4.6 | Leica | RRID:SCR_013673 | https://www.leica-microsystems.com/ |
| Other | Hypoxia chamber | STEMCELL Technologies | CAT# 27310 | https://www.stemcell.com/products/hypoxia-incubator-chamber.html |

## Bacterial strains

The following bacterial strains were used: *Escherichia coli* OP50, *E. coli* HT115(DE3), *E. coli* DH5α-GFP, *Pseudomonas aeruginosa* PA14, *P. aeruginosa* PA14-GFP, *P. aeruginosa* PA14 Δ*kinB*, *P. aeruginosa* PA14 Δ*lasR*, *P. aeruginosa* PA14 Δ*lysC*, *P. aeruginosa* PA14 Δ*rhlR*, *P. aeruginosa* PA14 Δ*phz* (lacking both the *phzA1-G1* and *phzA2-G2* operons), *P. aeruginosa* PA14 Δ*phzH*, *P. aeruginosa* PA14 Δ*phzM*, *P. aeruginosa* PA14 Δ*phzS*, and *P. aeruginosa* PA14 Δ*ptsP*. The cultures of these bacteria were grown in Luria-Bertani (LB) broth at 37°C.

## *C. elegans* strains and growth conditions

*C. elegans* hermaphrodites were maintained on *E. coli* OP50 at 20°C unless otherwise indicated. Bristol N2 was used as the wild-type control unless otherwise indicated. Strains FK181 *ksIs2 [daf-7p::GFP + rol-6(su1006)]*, DMS640 *nIs470 [cysl-2p::GFP + myo-2p::mCherry]*, CX4544 *ocr-2(ak47)*, JY190 *osm-9(yz6)*, JT23 *aex-5(sa23)*, MT1083 *egl-8(n488)*, JT307 *egl-9(sa307)*, DA609 *npr-1(ad609)*, and RB2302 *daf-7(ok3125)* were obtained from the Caenorhabditis Genetics Center (University of Minnesota, Minneapolis, MN). The *osm-9(yz6);egl-9(sa307)* and *daf-7(ok3125);npr-1(ad609)* double mutants were obtained by standard genetic crosses. The *daf-7(ok3125)* and *daf-7(ok3125);npr-1(ad609)* hermaphrodites were maintained on *E. coli* OP50 at 15°C.

## RNA interference (RNAi)

RNAi was used to generate loss-of-function RNAi phenotypes by feeding nematodes *E. coli* strain HT115(DE3) expressing double-stranded RNA (dsRNA) homologous to a target gene (**Fraser et al., 2000**; **Timmons and Fire, 1998**). RNAi was carried out as described previously (**Singh and Aballay, 2017**). Briefly, *E. coli* with the appropriate vectors were grown in LB broth containing ampicillin (100 μg/mL) and tetracycline (12.5 μg/mL) at 37°C overnight and plated onto NGM plates containing 100 μg/mL ampicillin and 3 mM isopropyl β-D-thiogalactoside (IPTG) (RNAi plates). RNAi-expressing bacteria were allowed to grow overnight at 37°C. Gravid adults were transferred to RNAi-expressing bacterial lawns and allowed to lay eggs for 2 hr. The gravid adults were removed, and the eggs were allowed to develop at 20°C to young adults for subsequent assays. The RNAi clones were from the Ahringer RNAi library.

## *P. aeruginosa* lawn avoidance assays

The bacterial cultures were grown by inoculating individual bacterial colonies into 2 mL of LB broth and growing them for 10–12 hr on a shaker at 37°C. Then, 20 μL of the culture was plated onto the center of 3.5-cm-diameter standard slow-killing (SK) plates (modified NGM agar plates (0.35% instead of 0.25% peptone)). The plates were then incubated under the following conditions: 37°C for 12 hr; 37°C for 24 hr; 37°C for 24 hr followed by 25°C for 24 hr; and 37°C for 24 hr followed by 25°C for 48 hr. The *P. aeruginosa* lawns obtained upon incubation at 37°C for 12 hr were used for avoidance assays unless otherwise indicated. Thirty synchronized young gravid adult hermaphroditic animals grown on *E. coli* HT115(DE3) containing control vector or an RNAi clone targeting a gene were transferred outside the indicated bacterial lawns, and the numbers of animals on and off the lawns were counted at the specified times for each experiment. Three 3.5-cm-diameter plates were used per trial in every experiment. The experiments were performed at 25°C. The percent occupancy was calculated as $(N_{on}\ lawn/N_{total}) \times 100$. At least three independent experiments were performed.

## *E. coli* lawn avoidance with purified phenazines

*E. coli* OP50 cultures were grown by inoculating individual bacterial colonies into 10 mL of LB broth and growing them for 10–12 hr on a shaker at 37°C. The cultures were concentrated 10–20-fold, and 20 μL was plated onto the center of 3.5-cm-diameter modified NGM agar plates and incubated at 37°C for 12 hr. The stock solutions of different phenazines, which were prepared in ethanol, were diluted to either 10 or 20 μg in M9 salt solution to a final volume of 20 μL and added onto the *E. coli* lawns. For control plates, the equivalent amount of ethanol was mixed with M9 salt solution and added onto the *E. coli* lawns. These plates were then incubated at room temperature for 30 min before seeding with 20 synchronized young gravid adult hermaphroditic animals. The experiments

were performed at 25°C. The percent occupancy was calculated as ($N_{on}$ lawn/$N_{total}$)×100. At least three independent experiments were performed.

## Two-choice preference assays

*P. aeruginosa* and *E. coli* HT115 cultures were grown by inoculating individual bacterial colonies into 2 mL and 10 mL of LB broth, respectively, and growing them for 10–12 hr on a shaker at 37°C. *E. coli* HT115 cultures were concentrated 10 to 20-fold before seeding on plates. Then, 20 μL of each inoculum was plated diagonally opposite onto 3.5-cm-diameter SK plates and incubated at 37°C for 12 hr. The plates were cooled to room temperature for at least 30 min before seeding with animals. Thirty synchronized young gravid adult hermaphroditic animals grown on *E. coli* HT115(DE3) containing control vector or an RNAi clone targeting a gene were transferred to the center of plates equidistant from both the lawns. For the experiments at 5% oxygen, the hypoxia chamber containing the plates was purged with 5% oxygen (balanced with nitrogen) for 5 min at a flow rate of 25 L/min. The chamber was then sealed and incubated at 25°C. For control, the two-choice assay plates were incubated at ambient oxygen. The numbers of animals on both lawns were counted at the specified times for each experiment. Three 3.5-cm-diameter plates were used per trial in every experiment. Experiments were performed at 25°C. The *P. aeruginosa* choice index (*P. aeruginosa* CI) was calculated as follows:

$$P.\ aeruginosa\ \mathrm{CI} = \frac{[(\text{No. of worms on } P.\ aeruginosa) - (\text{No. of worms on } E.\ coli)]}{[(\text{No. of worms on } P.\ aeruginosa) + (\text{No. of worms on } E.\ coli)]}$$

At least three independent experiments were performed.

## Intestinal colonization and bloating assay at 8% oxygen

*E. coli* DH5α-GFP cultures were grown by inoculating individual bacterial colonies into 25 mL of LB broth containing ampicillin (100 μg/mL) and growing them for 10–12 hr on a shaker at 37°C. The cultures were concentrated 10–20-fold, and 300 μL were plated onto the center of 6-cm-diameter NGM agar plates containing ampicillin (100 μg/mL) and incubated at 25°C for 48 hr. Fifty synchronized young gravid adult hermaphroditic N2 animals were transferred to plates containing *E. coli* DH5α-GFP lawns. The plates were then placed in a hypoxia chamber and the lids of the plates were removed. The hypoxia chamber was then purged with 8% oxygen (balanced with nitrogen) for 5 min at a flow rate of 25 L/min. The chamber was then sealed and incubated at 25°C for 24 hr. The control plates were incubated at ambient oxygen. After 24 hr of incubation, the animals were imaged for bacterial colonization and intestinal lumen diameter quantification.

## *C. elegans* killing assays on *Pseudomonas aeruginosa* PA14

The *C. elegans* killing assays were carried out on wild-type *P. aeruginosa* PA14 lawns that were incubated at 37°C for 12 hr, or 37°C for 24 hr followed by 25°C for 48 hr. The bacterial lawns were prepared as described above. For full-lawn killing assays, 20 μL of an overnight culture of *P. aeruginosa* PA14 variants grown at 37°C was spread on the complete surface of 3.5-cm-diameter SK plates. The plates were incubated at 37°C for 12 hr and then cooled to room temperature for at least 30 min before seeding with synchronized young gravid adult hermaphroditic animals. The killing assays were performed at 25°C, and live animals were transferred daily to fresh plates. Animals were scored at the times indicated and were considered dead when they failed to respond to touch. The killing assays with different *P. aeruginosa* PA14 mutants were carried out on full lawns.

## Quantification of intestinal bacterial loads

Intestinal bacterial loads were quantified as described previously (*Singh and Aballay, 2019a*). Briefly, *P. aeruginosa*-GFP lawns were prepared as described above. The plates were cooled to room temperature for at least 30 min before seeding with young gravid adult hermaphroditic animals. The assays were performed at 25°C. At the indicated times of exposure, the animals were transferred from *P. aeruginosa*-GFP plates to the center of fresh *E. coli* plates for 10 min to eliminate *P. aeruginosa*-GFP stuck to their body. Animals were transferred to the center of a new *E. coli* plate for 10 min to further eliminate external *P. aeruginosa*-GFP. Animals were transferred to fresh *E. coli* plates a third time for 10 min. Subsequently, ten animals/condition were transferred into 50 μL of

PBS plus 0.01% Triton X-100 and ground. Serial dilutions of the lysates ($10^1$, $10^2$, $10^3$, $10^4$) were seeded onto LB plates containing 50 µg/mL of kanamycin to select for *P. aeruginosa*-GFP cells and grown overnight at 37°C. Single colonies were counted the next day and represented as the number of bacterial cells or CFU per animal. Three independent experiments were performed for each condition.

### *daf-7p::GFP* induction assays

To measure the induction of *daf-7p::GFP* in the ASI and ASJ neuron pairs, 12 hr (incubated at 37°C for 12 hr) or 72 hr (incubated at 37°C for 24 hr followed by 25°C for 48 hr) lawns of wild-type *P. aeruginosa* PA14 were used. The bacterial lawns were prepared as described above. The plates were cooled to room temperature for at least 30 min before seeding with *daf-7p::GFP* reporter young gravid adult hermaphroditic animals. These plates were incubated at 25°C for the indicated times, and then the animals were prepared for fluorescence imaging. To measure the induction of *daf-7p::GFP* upon exposure to different *P. aeruginosa* PA14 mutants, the bacterial lawns were prepared by incubation at 37°C for 12 hr. Young gravid adult hermaphroditic *daf-7p::GFP* animals were transferred to these plates and incubated at 25°C for 4 hr before preparing the animals for fluorescence imaging. The fluorescence intensity in the ASI and ASJ neurons was quantified using Image J software.

### Fluorescence imaging

Fluorescence imaging was carried out as described previously (*Singh and Aballay, 2017*) with slight modifications. Briefly, the animals were anesthetized using an M9 salt solution containing 50 mM sodium azide and mounted onto 2% agar pads. The animals were then visualized using a Leica M165 FC fluorescence stereomicroscope.

### Quantification of intestinal lumen bloating

After the indicated treatment, the animals were anesthetized using an M9 salt solution containing 50 mM sodium azide, mounted onto 2% agar pads, and imaged. The diameter of the intestinal lumen was measured using the Leica LAS v4.6 software. At least 10 animals were used for each condition.

### *cysl-2p::GFP* induction assays

Synchronized young gravid adult hermaphroditic *cysl-2p::GFP* animals grown on *E. coli* HT115 were transferred onto *E. coli* HT115 and *P. aeruginosa* lawns. *P. aeruginosa* and *E. coli* HT115 cultures were grown by inoculating individual bacterial colonies into 2 mL and 10 mL of LB broth, respectively, and growing them for 8–10 hr on a shaker at 37°C. *E. coli* HT115 cultures were concentrated 10 to 20-fold before seeding on plates. Then, 20 µL of each inoculum was plated onto the center of 3.5-cm-diameter SK plates. For full lawns of *P. aeruginosa*, 20 µL of inoculum was spread to completely cover the surface of 3.5-cm-diameter SK plates. The plates were incubated at 37°C for 12 hr and then cooled to room temperature for at least 30 min before seeding with synchronized young gravid adult hermaphroditic *cysl-2p::GFP* animals. The COPAS Biosort machine (Union Biometrica) was used to measure the time of flight (length) and fluorescence of individual worms. At least 100 worms were measured for each condition.

### RNA isolation and quantitative reverse transcription-PCR (qRT-PCR)

Animals were synchronized by egg laying. Approximately 35 N2 gravid adult animals were transferred to 10 cm RNAi plates seeded with *E. coli* HT115 and allowed to lay eggs for 4 hr. The gravid adults were then removed, and the eggs were allowed to develop at 20°C. The animals were grown on the *E. coli* HT115 plates at 20°C until the young adult stage. Subsequently, the animals were transferred to 3.5-cm-diameter SK plates seeded with 20 µL of *P. aeruginosa* and pre-incubated at either 37°C for 12 hr or 37°C for 24 hr followed by 25°C for 48 hr. The control animals were maintained on *E. coli* HT115. After transfer of the animals, the plates were incubated at 25°C for 8 hr. Subsequently, the animals were collected, washed with M9 buffer, and frozen in TRIzol reagent (Life Technologies, Carlsbad, CA). Total RNA was extracted using the RNeasy Plus Universal Kit (Qiagen, Netherlands). Residual genomic DNA was removed using TURBO DNase (Life Technologies,

Carlsbad, CA). A total of 6 µg of total RNA was reverse-transcribed with random primers using the High-Capacity cDNA Reverse Transcription Kit (Applied Biosystems, Foster City, CA).

qRT-PCR was conducted using the Applied Biosystems One-Step Real-time PCR protocol using SYBR Green fluorescence (Applied Biosystems) on an Applied Biosystems 7900HT real-time PCR machine in 96-well-plate format. Twenty-five-microliter reactions were analyzed as outlined by the manufacturer (Applied Biosystems). The relative fold-changes of the transcripts were calculated using the comparative $CT (2^{-\Delta\Delta CT})$ method and normalized to pan-actin (act-1,–3, −4). The cycle thresholds of the amplification were determined using StepOnePlus software (Applied Biosystems). All samples were run in triplicate. The primer sequences have been described earlier (*Singh and Aballay, 2019a*).

## Quantification and statistical analysis

The statistical analysis was performed with Prism 8 (GraphPad). All error bars represent the standard deviation (SD). The two-sample t test was used when needed, and the data were judged to be statistically significant when $p < 0.05$. In the figures, asterisks (*) denote statistical significance as follows: *, $p < 0.05$, **, $p < 0.001$, ***, $p < 0.0001$, as compared with the appropriate controls. The Kaplan-Meier method was used to calculate the survival fractions, and statistical significance between survival curves was determined using the log-rank test. All experiments were performed in triplicate.

## Acknowledgements

We thank Anne Clatworthy from Deborah Hung's lab (Harvard Medical School) for providing the *P. aeruginosa* PA14 strains ΔkinB and ΔlasR, Glynis L Kolling from Jason A Papin's lab (University of Virginia) for providing the *P. aeruginosa* PA14 strain ΔlysC, Thomas K Wood (Pennsylvania State University) for providing the *P. aeruginosa* PA14 strain ΔrhlR, Lars Dietrich (Columbia University, NY) for providing the *P. aeruginosa* PA14 strains Δphz (lacking both the *phzA1-G1* and *phzA2-G2* operons), ΔphzH, ΔphzM and ΔphzS, and Meta Kuehn (Duke University Medical Center, NC) for providing the *P. aeruginosa* PA14 strain ΔptsP. Some strains used in this study were provided by the Caenorhabditis Genetics Center (CGC), which is funded by the NIH Office of Research Infrastructure Programs (P40OD010440).

## Additional information

### Funding

| Funder | Grant reference number | Author |
| --- | --- | --- |
| National Institute of General Medical Sciences | GM0709077 | Alejandro Aballay |
| National Institute of Allergy and Infectious Diseases | AI117911 | Alejandro Aballay |

The funders had no role in study design, data collection and interpretation, or the decision to submit the work for publication.

### Author contributions

Jogender Singh, Conceptualization, Data curation, Formal analysis, Investigation, Methodology, Writing—original draft, Writing—review and editing; Alejandro Aballay, Conceptualization, Data curation, Formal analysis, Funding acquisition, Writing—original draft, Project administration, Writing—review and editing

### Author ORCIDs

Alejandro Aballay https://orcid.org/0000-0002-5975-3352

### Decision letter and Author response

Decision letter https://doi.org/10.7554/eLife.50033.sa1
Author response https://doi.org/10.7554/eLife.50033.sa2

## Additional files

### Supplementary files
• Transparent reporting form

### Data availability
All data generated or analysed during this study are included in the manuscript and supporting files.

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
