## [Decision Letter]

**Acceptance summary:**

The manuscript challenges the current view that induced expression of *daf-7* in ASJ sensory neurons has the major role in transmitting the pathogen information for generation of the pathogen avoidance behavior, because the authors show that expression of *daf-7* in ASJ is not correlated with pathogen avoidance, but they instead show clearly that intestinal bloating is the major cause. Upon revision, authors addressed all major concerns that the reviewers raised. We are therefore glad to accept the manuscript for publication in *eLife*.

**Decision letter after peer review:**

Thank you for submitting your article "Intestinal infection regulates behavior and learning via neuroendocrine signaling" for consideration by *eLife*. Your article has been reviewed by three peer reviewers, and the evaluation has been overseen by a Reviewing Editor and K VijayRaghavan as the Senior Editor. The following individuals involved in review of your submission have agreed to reveal their identity: Javier Irazoqui (Reviewer #1).

The reviewers have discussed the reviews with one another and the Reviewing Editor has drafted this decision to help you prepare a revised submission.

All three reviewers appreciate the importance of the work, especially because it challenges the prevailing view of the major role of *daf-7* in ASJ in transmitting the pathogen information for generation of the pathgen avoidance behavior. However, they also agree that some of the authors' assertion needs to be substantiated further. We therefore invite revision of the manuscript, and as a consensus of the reviewers, we request the following to be addressed in revision.

Essential revisions:

1) The authors seem to claim that the pathogen avoidance behavior is driven by aerotaxis (at least in the present manuscript it sounds so). However, is it really aerotaxis which causes preference of *E. coli* over pathogenic bacteria after pathogenic exposure? The authors' claim is mainly based on the fact that lawn of *P. aeruginosa* has lower oxygen level than that of *E. coli*, and the requirement of *npr-1, ocr-2* and *osm-9* for avoidance and suppression by *egl-9*. However, these pathways regulate other functions as well. We request authors either test whether dead bacteria cannot cause the avoidance behavior, or perform behavioral assay that directly tests aerotaxis. Another option is to weaken the claim that it is aerotaxis.

2) Genetic requirements were assessed by RNAi for *nol-6, aex-5* and *egl-8*. Because RNAi does not always generate specific and complete loss of the targeted gene, and because mutants are available for all these genes, the results need to be backed up by using respective mutants.

3) The following point is not an absolute requirement, but it would be nice to be added: this study intriguingly shows that increased ASJ-expression of *daf-7* has no correlation with lawn avoiding. Since this finding will be very important for the field, it will be useful to provide a more complete analysis on *daf-7* by including ASI-expression of *daf-7*, because Meisel et al. shows that *daf-7* in ASI also rescues the lawn avoidance defects of *daf-7* mutant and the expression of *daf-7* in ASI also increases by exposure to *Pseudomonas*.

Reviewer #1:

In this very interesting paper, Singh and Aballay tackle a central question in innate immunity in general and in *C. elegans* innate immunity in particular: how is infection detected and host defense mechanisms engaged in vivo? The paradigm in this manuscript is *C. elegans* aversive learning, in which behavioral innate immunity is activated during infection so that the animal can escape an unfavorable (pathogenic) environment. The assays are very simple: in one case (lawn assay), the animals are exposed to stimuli (chemical, bacterial) on lawns of bacteria, and then their preference to stay on the lawn (and, thus, feed) or avoid it (and, thus, escape infection) is measured over time. In the other case (choice assay), animals reared on non-pathogenic *E. coli* are given the choice to continue eating *E. coli* or to feed on the "tastier" *Pseudomonas* aeruginosa – which is pathogenic.

First, the investigators dispel the notion, put forward in Meisel et al. (2014) in Cell, that sensory-neuron-mediated detection of phenazine-1-carboxamide is the trigger of aversive learning and pathogen avoidance through the induction of *daf-7* expression in ASJ neurons. This result is extremely important to the field, as many researchers have expressed their doubts about such a simplistic model, and is consistent with prior research from the Ausubel lab that showed that phenazine mutants of *P. aeruginosa* are just as virulent as wild type (replicated in this study).

Then, the investigators set out to determine what really triggers aversion during infection. First, they find that "older" solid cultures of *P. aeruginosa* are more virulent than "younger" ones. They establish that virulence correlates with intestinal distention, or "bloating". Drawing on their seminal work published earlier this year in Dev Cell, they find that older more pathogenic *Pseudomonas* induce gut-derived neuropeptides more strongly than younger ones, and that intestinal distention is required for aversive learning in the lawn assay. Then, they show that avirulent mutants of *Pseudomonas*, identified in a whole genome screen by the Ausubel lab, are defective in inducing aversive learning by the same assay. The nail in the phenazine/*daf-7* coffin is presented in Figure 4F,G, where they show a complete lack of correlation between *daf-7* ASJ expression and aversive learning, but almost perfect correlation with *Pseudomonas* virulence. After establishing that, they turn to preference. What makes *E. coli* less "tasty" than *Pseudomonas*, despite the virulence? Why does this choice invert after about 12 hours? To answer the first question, they examine aerotaxis. Consistent with the idea (and previous research) that *Pseudomonas* lawns exhibit lower oxygen tension (preferred by wild type *C. elegans*) than *E. coli* ones, they find induction of a HIF-1 target gene only in animals feeding on *Pseudomonas*, not *E. coli*. Then, they examine mutants. *daf-7* mutant exhibit no choice; *npr-1* mutants invert their choice more slowly than wild type. The double mutants show additive effects. Importantly, they also show that distention-resistant *nol-6* RNAi worms also do not invert their choice – they never learn. Conversely, *ocr-2* and *osm-9* mutants, which prefer higher oxygen tensions, switch their preference to *E. coli* and learn to avoid *Pseudomonas* lawns faster than wild type; this is suppressed by loss of *egl-9*, a negative regulator of HIF-1. The implication, which is not substantiated or tested, is that HIF-1 activation and thus target gene transcription is epistatic to the loss of aerotaxis behavior phenotype in *ocr-2* or *osm-9* mutants.

In summary, this important study corrects the record on an important subject, and identifies the genetics in host and pathogen that are involved in aversive learning and food choice during infection.

Reviewer #2:

Singh and Aballay examine the mechanisms that regulate the avoidance of pathogenic *P. aeruginosa* bacteria in *C. elegans*.

They first test the role of several phenazines in eliciting avoidance and increasing *daf-7* expression in ASJ. They report that only one type of phenazine increases *daf-7* expression in ASJ, but it does not generate avoidance when put on *E. coli*. They also find that another type of phenazine repels worms although it does not increase ASJ expression of *daf-7* and mutants lacking phenazines repel worms similarly as wild type. These results show that phenazines are not required for the repulsion of the worms.

Next, they examine whether the growth conditions regulate lawn repulsion by using 72 hour lawns and 12 hour lawns. They found that 72 hour lawns that strongly repel worms cause intestine bloating but 12 hour lawns that repel worms less do not cause bloating. Meanwhile, the 72 hour lawns, but not the 12 hour lawns, also regulate transcription similarly as intestine bloating and decreasing bloating by *nol-6* RNAi delays avoidance. Meanwhile, they show that enhancing intestine bloating (*aex-5, egl-8* RNAi) enhances repulsion. These results demonstrate a strong link between intestine bloating and lawn avoidance.

Finally, they find that several *P. aeruginosa* mutants with weaker virulence show decreased repulsion to the worms and mutating aerotaxis gene *oms-9* and *ocr-2*, alters the switch of preference from pathogenic *P. aeruginosa* to *E. coli*.

The findings in this study demonstrate a strong link between intestine bloating and lawn avoidance and provide a new mechanism for the worms to avoid infectious bacteria. Several questions need to be addressed.

1) The results and the analyses using RNAi should be supported by genetic experiments.

2) Previous studies show that exposure to the pathogenic *P. aeruginosa* bacteria increases expression of *daf-7* in both ASJ and ASI and that *daf-7* expression in either neuron rescues the lawn leaving phenotype in *daf-7*. While this work tests the association between ASJ expression of *daf-7* and lawn leaving behavior, it will be helpful in furthering the understanding of *daf-7* function by examining ASI expression of *daf-7*.

3) The manuscript states that the findings on ASJ expression of *daf-7* argue against the message provided by a previous study (Meisel et al., 2014). This is confusing, because Meisel et al. does not seem to propose that increased expression of *daf-7* in ASJ, induced by pathogenic *P. aeruginosa* or phenazine-1-carboxamide and pyochelin, elicits avoidance. Rather, they propose that it is one of sensory responses to the bacteria (for example, they conclude that "Our findings demonstrate how specific bacteria can exert effects on host behavior and physiology and point to how secondary metabolites may serve as environmental cues that contribute to pathogen discrimination and avoidance."). It is important to clarify these issues for the readers to better understand the complexity of the question.

Reviewer #3

In "Intestinal infection regulates behavior and learning via neuroendocrine signaling," the authors show that feedback from the intestine modulates the worm behavioral response to infection with *Pseudomonas aeruginosa* – namely pathogen avoidance. The response requires *daf-7* and *npr-1*; loss of either of these genes reduces pathogen avoidance and loss of both completely blocks the process. It had previously been thought that *daf-7* expression in the ASJ neurons might drive this process because a *daf-7p::GFP* reporter construct is activated in these neurons upon exposure to *P. aeruginosa*. However, the authors show that the behavior is not correlated with this expression. In one example, 12 hour and 72 hour lawns of *P. aeruginosa* cause equal amounts of *daf-7p::GFP* expression, but elicit different levels of lawn avoidance. In a second example, worms are exposed to mutants of *P. aeruginosa* that change how avidly they avoid the lawns, but the degree of avoidance is not correlated with the levels of induced ASJ *daf-7p::GFP* expression. Rather, the authors argue that the level of intestinal bloating caused by the *P. aeruginosa* infection drives the pathogen avoidance response. Several examples of correlation are given.

1) The phenazine 1-hydroxyphenazine is the only one of four tested that causes both bloating and avoidance.

2) A 72 hour lawn causes more rapid bloating and more rapid avoidance.

3) The degree of bloating caused by different *P. aeruginosa* mutants is correlated with the degree of avoidance.

4) *C. elegans* mutants that have more bloating have more avoidance, and those that have less bloating have less avoidance. The data in total supports in a correlative manner the authors' postulate that the changes in the intestine (bloating) causes the ensuing pathogen avoidance behavior.

The authors then argue that it is aerotaxis behavior via neuroendocrine signaling that drives the learned pathogen avoidance behavior. Worms like bacterial lawns with less oxygen and *P. aeruginosa* lawns are known to have less oxygen. However, following intestinal bloating and activation of the NPR-1/DAF-7 neuroendocrine pathway on *P. aeruginosa*, the animals come to prefer the higher oxygen lawns of *E. coli*. The argument is supported by the fact that animals mutant in *daf-7* and/or *npr-1* cannot learn to avoid *P. aeruginosa* lawns.

Overall, this is a very interesting study and it will be of broad interest to the worm community. However, I am concerned about the heavy dependence on correlative results on which the main conclusions rest.

1) Are oxygen levels really the signal the worms learn to recognize as a sign of a pathogenic lawn or is it a correlation? Is there a way to equalize the oxygen levels between the pathogenic and non-pathogenic lawns? Then one could see if the worms can still learn to avoid the lawn (and if intestinal bloating still occurs (see point 2)). Alternatively, maybe the animals trained to avoid *P. aeruginosa* could be tested in an assay designed to test their aerotaxis preferences?

2) The relationship, if any, between low/high oxygen and intestinal bloating is not clear. Does the low oxygen cause the bloating on *P. aeruginosa*, or could it? Or do the authors think that the high/low oxygen is just how the worms learn to distinguish between a good (*E. coli*) and bad (*P. aeruginosa*) lawn?

3) Worms like bacterial lawns with less oxygen and *P. aeruginosa* lawns are known to have less oxygen. Do the authors think this explains the initial attraction to the *P. aeruginosa* lawn?

---

## [Author Response]

Essential revisions:1) The authors seem to claim that the pathogen avoidance behavior is driven by aerotaxis (at least in the present manuscript it sounds so). However, is it really aerotaxis which causes preference of E. coli over pathogenic bacteria after pathogenic exposure? The authors' claim is mainly based on the fact that lawn of P. aeruginosa has lower oxygen level than that of E. coli, and the requirement of npr-1, ocr-2 and osm-9 for avoidance and suppression by egl-9. However, these pathways regulate other functions as well. We request authors either test whether dead bacteria cannot cause the avoidance behavior, or perform behavioral assay that directly tests aerotaxis. Another option is to weaken the claim that it is aerotaxis.

We have both carried out additional experiments to directly study the role of aerotaxis in the change of microbial preference and discussed the possibility of additional cues, other than aerotaxis. As shown in new Figure 6—figure supplement 2, we have provided further evidence for the role of aerotaxis in the change of microbial preference. The revised manuscript reads: “We were also able to elicit a rapid preference towards *E. coli* by exposing the animals to 5% oxygen (Figure 6—figure supplement 2).”

In addition, we have also added that our studies do not rule out the role of additional cues in the elicitation of the behavioral changes: “We cannot rule out that other metabolites or virulence factors may play a role in the elicitation of pathogen avoidance.”

2) Genetic requirements were assessed by RNAi for nol-6, aex-5 and egl-8. Because RNAi does not always generate specific and complete loss of the targeted gene, and because mutants are available for all these genes, the results need to be backed up by using respective mutants.

We have added data with *aex-5* and *egl-8* mutants in new Figure 6H. The *nol-6* mutant animals are temperature-sensitive sterile. Therefore, we have used RNAi for knocking down *nol-6*. Previous studies have shown that knockdown of *nol-6* by RNAi reduces bacterial colonization and intestinal bloating (Fuhrman et al., 2009). In our current study, we also confirmed that *nol-6* RNAi reduces intestinal bloating (Figure 3F,G, Figure 3—figure supplement 1B,C). Since we are testing the role of intestinal bloating (and not the function of *nol-6* per se) in the avoidance behavior, we believe that the use of *nol-6* RNAi is reliable and sufficient.

3) The following point is not an absolute requirement, but it would be nice to be added: this study intriguingly shows that increased ASJ-expression of daf-7 has no correlation with lawn avoiding. Since this finding will be very important for the field, it will be useful to provide a more complete analysis on daf-7 by including ASI-expression of daf-7, because Meisel et al. shows that daf-7 in ASI also rescues the lawn avoidance defects of daf-7 mutant and the expression of daf-7 in ASI also increases by exposure to Pseudomonas.

We have added data on *daf-7* expression in ASI neurons. The data show that increased ASI expression of *daf-7* does not have a correlation with lawn avoidance either. The data is presented in new Figure 2—figure supplement 1, Figure 3—figure supplement 1D, and Figure 4—figure supplement 1B.

Reviewer #1:[…]In summary, this important study corrects the record on an important subject, and identifies the genetics in host and pathogen that are involved in aversive learning and food choice during infection.

We thank the reviewer for a very good summary of our work and for highlighting the importance of the findings.

Reviewer #2:[…]The findings in this study demonstrate a strong link between intestine bloating and lawn avoidance and provide a new mechanism for the worms to avoid infectious bacteria. Several questions need to be addressed.1) The results and the analyses using RNAi should be supported by genetic experiments.

We have added data with *aex-5* and *egl-8* mutants in new Figure 6H. The *nol-6* mutant animals are temperature-sensitive sterile. Therefore, we used RNAi for knocking down *nol-6*. Previous studies have shown that knockdown of *nol-6* by RNAi reduces bacterial colonization and intestinal bloating (Fuhrman et al., 2009). In our current study, we also showed that *nol-6* RNAi reduces intestinal bloating (Figure 3F,G, Figure 3—figure supplement 1B,C). Since we are testing the role of intestinal bloating (and not the function of *nol-6* per se) in the avoidance behavior, we believe that the data with *nol-6* RNAi is sufficient.

2) Previous studies show that exposure to the pathogenic P. aeruginosa bacteria increases expression of daf-7 in both ASJ and ASI and that daf-7 expression in either neuron rescues the lawn leaving phenotype in daf-7. While this work tests the association between ASJ expression of daf-7 and lawn leaving behavior, it will be helpful in furthering the understanding of daf-7 function by examining ASI expression of daf-7.

We have added data on *daf-7* expression in ASI neurons. The data show that increased ASI expression of *daf-7* does not have correlation with lawn avoidance either. The data is presented in Figure 2—figure supplement 1, Figure 3—figure supplement 1D, and Figure 4—figure supplement 1B.

3) The manuscript states that the findings on ASJ expression of daf-7 argue against the message provided by a previous study (Meisel et al., 2014). This is confusing, because Meisel et al. does not seem to propose that increased expression of daf-7 in ASJ, induced by pathogenic P. aeruginosa or phenazine-1-carboxamide and pyochelin, elicits avoidance. Rather, they propose that it is one of sensory responses to the bacteria (for example, they conclude that "Our findings demonstrate how specific bacteria can exert effects on host behavior and physiology and point to how secondary metabolites may serve as environmental cues that contribute to pathogen discrimination and avoidance."). It is important to clarify these issues for the readers to better understand the complexity of the question.

We agree with the reviewer that this is a complex question. However, we would like to bring out that Meisel et al. did indeed suggest that induction of DAF-7 in the ASJ neuron pair promotes the avoidance behavior. Meisel et al., 2014 reads “Secondary metabolites phenazine-1-carboxamide and pyochelin activate a G-protein-signaling pathway in the ASJ chemosensory neuron pair that induces expression of the neuromodulator DAF-7/TGF-β. DAF-7, in turn, activates a canonical TGF-β signaling pathway in adjacent interneurons to modulate aerotaxis behavior and promote avoidance of pathogenic *P. aeruginosa*.”

Several studies have referred to Meisel et al., 2014, highlighting the role of induction of DAF-7 in ASJ in regulation of pathogen avoidance behavior, including: Hilbert and Kim, *eLife*, 2017; Horspool and Chang, Sci Rep., 2017 (PMID: 28322326); Hao et al., *eLife*, 2018; and Harris et al., PloS Genetics, 2019 (PMID: 30849079).

From the above examples, it is clear that the notion in the field is that induction of DAF-7 in the ASJ neurons drives the *P. aeruginosa* avoidance behavior.

Reviewer #3[…]1) Are oxygen levels really the signal the worms learn to recognize as a sign of a pathogenic lawn or is it a correlation? Is there a way to equalize the oxygen levels between the pathogenic and non-pathogenic lawns? Then one could see if the worms can still learn to avoid the lawn (and if intestinal bloating still occurs (see point 2)). Alternatively, maybe the animals trained to avoid P. aeruginosa could be tested in an assay designed to test their aerotaxis preferences?

We have provided additional support that aerotaxis is important for the change in bacterial preference (Figure 6—figure supplement 2). The revised manuscript reads: “We were also able to elicit a rapid preference towards *E. coli* by exposing the animals to 5% oxygen (Figure 6—figure supplement 2).”

2) The relationship, if any, between low/high oxygen and intestinal bloating is not clear. Does the low oxygen cause the bloating on P. aeruginosa, or could it? Or do the authors think that the high/low oxygen is just how the worms learn to distinguish between a good (E. coli) and bad (P. aeruginosa) lawn?

We thank the reviewer for asking this intriguing question. To ask if low oxygen levels could be responsible for intestinal colonization and bloating, we incubated N2 animals on *E. coli*-GFP at 8% oxygen levels for 24 hours (*P. aeruginosa* lawns are known to have 8-10% oxygen: Reddy et al., 2011). As shown in new Figure 6—figure supplement 3, low oxygen levels per seare not responsible for intestinal bacterial colonization and bloating.

3) Worms like bacterial lawns with less oxygen and P. aeruginosa lawns are known to have less oxygen. Do the authors think this explains the initial attraction to the P. aeruginosa lawn?

It is hard to predict if the initial attraction towards *P. aeruginosa* is solely because of relatively lower oxygen levels. Since *C. elegans* behavior is modulated by both bacterial metabolites (attractants) and oxygen levels, it is likely that the initial attraction is a combination of multiple cues.